# ANYTIME NEURAL ARCHITECTURE SEARCH ON TABULAR DATA

## ABSTRACT

The increasing demand for tabular data analysis calls for transitioning from manual architecture design to Neural Architecture Search (NAS). This transition demands an efficient and responsive *anytime NAS* approach that is capable of returning current optimal architectures within any given time budget while progressively enhancing architecture quality with increased budget allocation. However, the area of research on Anytime NAS for tabular data remains unexplored. To this end, we introduce ATLAS, the first anytime NAS approach tailored for tabular data. ATLAS introduces a novel two-phase filtering-and-refinement optimization scheme with joint optimization, combining the strengths of both paradigms of training-free and training-based architecture evaluation. Specifically, in the filtering phase, ATLAS employs a new training-free architecture evaluation metric specifically designed for tabular data to efficiently estimate the performance of candidate architectures, thereby obtaining a set of promising architectures. Subsequently, in the refinement phase, ATLAS leverages a fixed-budget search algorithm to schedule the training of the promising candidates, so as to accurately identify the optimal architecture. To jointly optimize the two phases for anytime NAS, we also devise a budget-aware coordinator that delivers high NAS performance within constraints. Experimental evaluations demonstrate that our ATLAS can obtain a good performing architecture within any predefined time budget and return better architectures as and when a new time budget is made available. Overall, it reduces the search time on tabular data by up to 82.75x compared to existing NAS approaches.

## 1 INTRODUCTION

Tabular data analysis is increasingly pivotal in both industry and academia, supporting daily decision-making processes in real-world applications such as click-through rate prediction, online recommendation, and readmission prediction Gorishniy et al. (2022); Arik & Pfister (2021). In particular, Deep Neural Networks (DNNs) have surpassed the performance of traditional tree-based approaches in various applications Borisov et al. (2022); Cai et al. (2021), highlighting the increased need to design better-performing DNNs for tabular data.

Much of the research to date has been focusing on manually designing architectures following certain prior assumptions of tabular data Qin et al. (2021); Gorishniy et al. (2021); Xie et al. (2021); Popov et al. (2019); Levin et al. (2023); Chen et al. (2023a). Nonetheless, designing DNNs in such a trial-and-error manner is both labor/computation-intensive and time-consuming. In recent years, Neural Architecture Search (NAS) has been widely adopted to automate the architecture design for other data types such as images, texts, and videos White et al. (2023); Shala et al. (2023); Lee et al. (2023); Kadra et al. (2021); Wang et al. (2020b); Liu et al. (2019a), which employs various algorithms to search for more efficient and effective DNN architectures. As illustrated in Figure 1, a typical NAS approach Wistuba et al. (2019); Ren et al. (2021) comprises three integral components: a search space, a search strategy, and an architecture evaluation component. The search space defines the construction options for candidate architectures and the search strategy determines the architecture exploration method within the search space. Finally, the architecture evaluation component evaluates the performance of the architecture.

As with the growing demand for more effective and efficient NAS Yang et al. (2023); Zhang et al. (2020); Zhao et al. (2022), an increasing number of applications also require a response-time-aware

and cost-efficient NAS Demirovic et al. (2023); Li et al. (2023); Bohdal et al. (2023). Notably, small businesses and individual users tend to utilize cloud resources to support NAS goo (2023). To manage costs, these NAS users often need to allocate a predefined time budget for using cloud resources. In light of this, we advocate the concept of *Anytime Neural Architecture Search* (Anytime NAS), which refers to NAS approaches that can produce a suboptimal architecture within any given time budget and find a higher-quality architecture as more time budget become available.

The development of an Anytime NAS approach for tabular data presents several key challenges. First, the lack of NAS benchmark datasets for tabular data impedes the evaluation of NAS approaches, which requires repeated and expensive architecture evaluations Ying et al. (2019); Dong & Yang (2020). Second, while training-based NAS approaches can accurately evaluate architecture performance, they need to laboriously train numerous candidate architectures, each using hundreds to thousands of iterations Zoph & Le (2017). This computation-intensive process makes the existing NAS approaches slow and not *anytime*-capable. Recently, several NAS approaches on vision tasks Krishnakumar et al. (2022); Chen et al. (2021); Shu et al. (2022a); Lee et al. (2019); Tanaka et al. (2020) propose to reduce architecture evaluation costs via **T**raining f**R**ee **A**rch**I**tecture eva**L**uation m**Et****R**ics (TRAILERs), which quickly estimate the architecture performance by calculating certain architecture statistics without expensive full training. However, to the best of our knowledge, there is no research to date that investigates the effectiveness of such training-free evaluation on tabular data, and no TRAILER has been specifically designed for tabular data. Further, existing NAS approaches only introduce TRAILERs to enhance the conventional training-based iterative search process, e.g., for pretraining the search strategy or proposing candidate architectures for training-based evaluation Abdelfattah et al. (2021); Shu et al. (2022a); Lee et al. (2019). How to combine the strengths of both training-based and training-free paradigms in a more principled way and ensure improved architectures for an increased time budget remains unresolved.

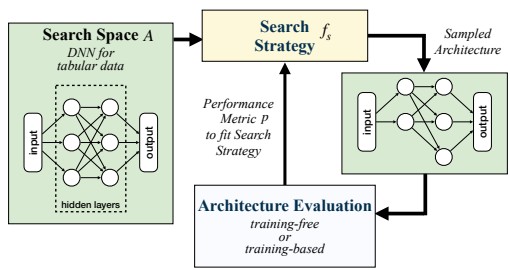

Figure 1: The Overview of NAS.

To address the above challenges, we propose ATLAS, an **A**ny**T**ime neura**L** **A**rchitecture **S**earch that is tailored for tabular data. We first construct a comprehensive NAS tabular data benchmark, which comprises more than 160,000 unique architectures over three real-world datasets. Using this benchmark, we conduct an extensive analysis of nine state-of-the-art TRAILERs, evaluating their performance in the context of tabular data. Second, we propose a new metric *ExpressFlow* based on our in-depth theoretical analysis, which effectively evaluates architecture performance while maintaining efficiency for tabular data applications. Third, to support anytime NAS, we propose to restructure the conventional NAS process into two decoupled phases, i.e., the filtering phase and the refinement phase, with joint optimization via a budget-aware coordinator. The filtering phase efficiently explores a large set of candidate architectures using our proposed ExpressFlow to obtain a set of promising architectures. Subsequently, the refinement phase employs the more expensive training-based architecture evaluation to accurately identify the best-performing architecture from the promising architectures. Following this two-phase optimization scheme, we propose our novel anytime NAS approach ATLAS that can identify a high-performing architecture within any given time budget and continue to refine the search results as more resources become available.

In summary, this paper makes the following contributions:

- We construct a comprehensive search space tailored for tabular data, termed *NAS-Bench-Tabular*, which comprises an extensive collection of 160,000 architectures with detailed training and evaluation statistics for benchmarking various NAS approaches on tabular data.

- We perform theoretical analysis and benchmark state-of-the-art TRAILERs initially proposed for vision tasks on real-world tabular datasets, and introduce the first TRAILER tailored for tabular data, ExpressFlow, which characterizes both the expressivity and trainability of architectures for more effective architecture evaluation than existing metrics.

- We propose ATLAS, the first NAS approach supporting anytime NAS on tabular data, which introduces a novel two-phase optimization scheme that combines the benefits of both training-free and training-based architecture evaluation.

- We demonstrate that our ATLAS empirically outperforms existing NAS approaches by a wide margin, reducing the architecture search time by up to $82.75$x, and achieves anytime NAS on tabular data.

The remainder of this paper is organized as follows: Section 2 presents the notations and terminology used, Section 3 describes the methodology, and Section 4 provides experimental results. Related work is summarized in Appendix A, and Section 5 concludes this paper.

## 2 NOTATION AND TERMINOLOGY

**Search Space**, denoted as $\mathcal{A}$, refers to a collection of possible architectures $\mathcal{A} = \{a\}$, each of which has a unique topology. Existing studies show that deep neural networks (DNNs) can already achieve state-of-the-art performance on tabular data, and the main technical challenge is to configure DNNs with the right number of hidden neurons for each layer, namely the layer sizes Cai et al. (2021); Yang et al. (2022); Levin et al. (2023); Gorishniy et al. (2021). Therefore, we employ DNNs as the backbone to construct the search space for tabular data. Specifically, for a DNN with $L$ layers and a candidate set of layer sizes $\mathcal{H}$, the search objective is to determine layer sizes for the $L$ hidden layers from $\mathcal{H}$, and the number of candidate architectures is $|\mathcal{H}|^L$. For example, a search space $\mathcal{A}$ defined by $L=6$ and $\mathcal{H}=\{8, 16, 32, 64, 128\}$ contains $5^6=15,625$ candidate architectures.

**Search Strategy** is responsible for proposing a candidate architecture $a_{i+1}$ for evaluation from the search space, denoted as $a_{i+1} = f_s(\mathcal{A}, \mathcal{S}_i)$, where $\mathcal{S}_i$ represents the state of the search strategy at the $i$-th iteration. The objective of the search strategy is to efficiently explore the search space by evaluating promising architectures Bohdal et al. (2023); Cai et al. (2020); Yang et al. (2022). Popular search strategies include random sampling Bergstra & Bengio (2012), reinforcement learning Zoph & Le (2017), evolutionary algorithm Real et al. (2019a), and Bayesian Optimization with HyperBand Falkner et al. (2018), and etc.

**Architecture Evaluation** refers to the assessment of architecture performance, and can be either training-based Zoph & Le (2017) or training-free Mellor et al. (2021); Li et al. (2023); Shu et al. (2022a); Tanaka et al. (2020). The performance obtained by the training-based architecture evaluation approaches is accurate, which however requires costly training. In contrast, training-free evaluation efficiently estimates architecture performance by computing certain architecture statistics using only a small batch of data for efficiency. Given an architecture $a$ parameterized by $\boldsymbol{\theta}$ and a batch of $B$ data samples $X_B$, a TRAILER computes a score $s_a$ to quantify the performance $p$, formally described as $s_a = \rho(a, \boldsymbol{\theta}, X_B)$, where $\rho(\cdot)$ is the *assessment function* of the TRAILER. A summary of all notations and terminologies in the paper can be found in Appendix H.

## 3 METHODOLOGY

Unlike conventional NAS approaches, our ATLAS is structured into two distinct phases: the *filtering phase* and the *refinement phase*, based on training-free and training-based architecture evaluation respectively, and optimized jointly via a budget-aware coordinator to support anytime NAS. In the filtering phase, ATLAS efficiently explores the search space, directed by a search strategy using our new TRAILER. Next, in the refinement phase, ATLAS evaluates the most promising architectures accurately via training-based evaluation. A coordinator is also introduced to guide the two phases, ensuring that ATLAS can deliver a high-performing architecture given a specified time budget $T_{max}$.

To ensure fair and consistent benchmarking for different NAS approaches, we first build NAS-Bench-Tabular on real-world tabular datasets in Section 3.1. Next, we characterize TRAILERs on two key properties, trainability and expressivity, and propose a more effective training-free metric for tabular data to accelerate the filtering phase in Section 3.2. Then, we employ a scheduling algorithm for the training-based architecture evaluation to optimize the refinement phase in Section 3.3. Finally, we introduce the budget-aware coordinator for the two phases to support anytime NAS in Section 3.4.

### 3.1 SEARCH SPACE DESIGN AND NAS-BENCH-TABULAR

Following prior studies Yang et al. (2022); Gorishniy et al. (2021); Chen et al. (2023b); Cai et al. (2021), the search space is set to a DNN backbone with $L$ layers and a set of candidate layer sizes $\mathcal{H}$. Each DNN layer comprises a linear transformation and a batch normalization layer, followed by a ReLU activation function. To construct a comprehensive search space for benchmarking, we establish $L=4$ and $|\mathcal{H}|=20$ with candidate layer sizes ranging from 8 to 512, resulting in a total of $20^4=160,000$ unique architectures. The architecture search is then to determine the sizes of each of the $L$ layers, which can be encoded with $L$ integers, denoting the number of neurons in corresponding layers.

The training procedure and hyperparameters are critical for NAS benchmarks, which can significantly affect the architecture evaluation results Ying et al. (2019); Dong & Yang (2020). Therefore, we examine key training hyperparameters for each benchmark dataset, including training iterations, training epochs, and the batch size, and then select the best hyperparameters for each dataset and conduct the full training of all candidate architectures on respective datasets.

We adopt three widely benchmarked tabular datasets Frappe, Diabetes, and Criteo Luo et al. (2023); Yang et al. (2022), and record five performance indicators for each architecture evaluated on respective datasets, i.e., training area under the curve (AUC), validation AUC, training time, training loss, and validation loss. With these performance indicators, NAS approaches can directly query the performance of each architecture without performing expensive training. More details, analysis, and discussions of NAS-Bench-Tabular are provided in Section 4.1 and Appendix C.

### 3.2 ARCHITECTURE FILTERING VIA TRAINING-FREE ARCHITECTURE EVALUATION

TRAILERs are designed to efficiently estimate the architecture performance using only a batch of data. Theoretically, TRAILERs characterize two key properties of the architecture that are related to its performance: *trainability* Wang & Fu (2023); Shin & Karniadakis (2020); Chen et al. (2021) and *expressivity* Hornik et al. (1989); Wang et al. (2023b); Raghu et al. (2017).

**Trainability** quantifies the degree to which the architecture can be effectively optimized through gradient descent, and the trainability of an architecture at initialization is crucial for the final performance. Particularly, parameters in DNNs store task-specific information, and their importance in learning the task significantly influences the architecture's performance. Therefore, an effective characterization of an architecture's trainability necessitates aggregating the importance of individual parameters. Notably, *synaptic saliency* Tanaka et al. (2020); Lee et al. (2019) quantifies the importance of respective parameters by $\Phi(\boldsymbol{\theta}) = f(\frac{\partial \mathcal{L}}{\partial \boldsymbol{\theta}}) \odot g(\boldsymbol{\theta})$, where $\mathcal{L}$ represents the loss function and $\odot$ denotes the Hadamard product. Typically, the trainability of an architecture can be quantified by aggregating the parameter importance: $\sum_{\boldsymbol{\theta}_i} \Phi(\boldsymbol{\theta}_i)$ Tanaka et al. (2020); Mellor et al. (2021).

**Expressivity** refers to the complexity of the function that the architecture can represent, which is strongly correlated with architecture's performance. Expressivity relies heavily on the architecture topology in terms of depth (how many layers) and width (number of neurons in a layer). Typically, wider and deeper architectures represent more complex functions Zhang et al. (2021); Hanin & Rolnick (2019). Further, parameters of the lower layers of architecture have more influence on the expressivity. This observation can be effectively quantified by trajectory length $\ell(\cdot)$ Raghu et al. (2017), which increases exponentially with the depth of the architecture. Specifically, for the $l$-th layer, $\ell(z^l(t)) = \int_t ||\frac{dz^l(t)}{dt}||$, where $z^l(t)$ is the trajectory of the $l$-th layer and $t$ is a scalar to parameterize it. For simplicity, we denote $z^l(t)$ as $z^l$. We provide more discussions of trainability and expressivity in Appendix A.3. Analysis and comparisons of existing TRAILERs are in Appendix G.1.

**ExpressFlow for enabling efficient architecture filtering for tabular data.** To characterize the architecture performance on both trainability and expressivity, we propose ExpressFlow which is tailored for DNNs on tabular data. ExpressFlow is based on *neuron saliency*, a more effective characterization of architecture performance. For the $n$-th neuron in the DNN, we quantify its saliency in the architecture, denoted as $\nu_n$. Specifically, this is computed as the product of the absolute value of the derivative of $\mathcal{L}$ with respect to the activated output of the neuron $z_n$, and the value of $z_n$ itself, i.e., $\nu_n = |\frac{\partial \mathcal{L}}{\partial z_n}| \odot z_n$, where $z_n = \sigma(\mathbf{w}\mathbf{x} + b)$, and $\mathbf{w}$ represents the incoming weights of the neuron, $\mathbf{x}$ is the neuron inputs, $b$ is the bias, and $\sigma$ is the activation function. Notably, for the ReLU activation function, $\nu_n = |\frac{\partial \mathcal{L}}{\partial z_n}| \odot z_n$ if $z_n > 0$, otherwise $\nu_n = 0$.

Importantly, not all neurons contribute equally. Their respective depths and widths within the architecture play critical roles in determining the architecture's performance. As supported by Raghu et al. (2017), we note that parameters at lower layers generally have a more significant impact on the performance of the architecture. To account for this, we recalibrate $\nu_n$ for each neuron at the layer $l$ inversely proportional to the trajectory length of that layer, namely $1/\ell(z^l)$. As $\ell(z^l)$ grows with layer depth, this recalibration highlights the significance of neurons in the lower layers. Also, following Lu et al. (2017); Zhang et al. (2021), to compensate for the varying impacts of layer width on neuron saliency, we further recalibrate $\nu_n$ for each neuron at layer $l$ in relation to the layer width, i.e. $\mathcal{K}_l$. Altogether, the recalibration weight for neuron saliency $\nu_n$ of neuron at layer $l$ is $\frac{\mathcal{K}_l}{\ell(z^l)}$. Then, we can perform a weighted aggregation of neuron saliency to derive a score that depicts the performance of the architecture $a$ on a batch of data $X_B$, as outlined below:

$$s_a = \rho_{ExpressFlow}(a) = \sum_{i=1}^{B}\sum_{n=1}^{N}\frac{\mathcal{K}_l}{\ell(z^l)}\nu_{in} = \sum_{i=1}^{B}\sum_{l=1}^{L}\frac{\mathcal{K}_l}{\ell(z^l)}(\sum_{n=1}^{N_l}|\frac{\partial\mathcal{L}}{\partial z_{in}}|\bigodot z_{in}) \qquad (1)$$

where $\nu_{in}$ and $z_{in}$ is the neuron saliency and activated output of $n$-th neuron computed on $i$-th sample, respectively. $N$ denotes the total number of neurons in the entire architecture, while $N_l$ represents the number of neurons in the $l$-th layer.

ExpressFlow is specifically designed to accommodate both trainability and expressivity. First, neuron saliency considers the activation value of neurons, i.e., the effects of neurons, which are the basic units to extract features in a DNN Levin et al. (2023); Cai et al. (2021). By computing the activated neuron output $z_n$, neuron saliency captures the complex and non-intuitive relationships among input features in tabular data. Second, neuron saliency calculates the derivatives of neurons. A larger gradient of the loss with respect to the activation $z_n$, i.e., $\frac{\partial\mathcal{L}}{\partial z_n}$, indicates higher importance of the features extracted by this neuron, and therefore, demonstrates the greater significance of the neuron to the prediction task. Last, neuron saliency's recalibration weight, i.e., $\frac{\mathcal{K}_l}{\ell(z^l)}$, is determined by the depth and width of its layer. Neurons in lower and wider layers receive higher saliency values, highlighting their larger influence on architecture performance. More in-depth theoretical analysis of ExpressFlow regarding its trainability and expressivity is provided in Appendix G.3.

**Architecture Filtering via ExpressFlow.** ExpressFlow of a given architecture can be computed within seconds and thus is highly efficient. Nevertheless, calculating ExpressFlow for all candidate architectures to select the highest-scored architecture is computationally intensive, especially for large search spaces. Therefore, a search strategy is required to further improve search efficiency, which guides the search for architectures of higher ExpressFlow scores. To this end, we leverage Regularized Evolution (RE) Real et al. (2019b), which maintains a diverse architecture population and executes iterative architecture sampling, architecture evaluation (using training-free ExpressFlow), and mutation (based on the current highest-scored architecture). RE offers several advantages. First, its gradient-free computation delivers high efficiency, which ensures rapid exploration and is well suited for anytime NAS on tabular data. Second, its aging mechanism facilitates diversified exploration of architectures, preventing the search from trapping in local optima. Last, its fine-grained mutation makes sure that promising candidate architectures within a local search space are all explored and recorded. Therefore, with RE as the search strategy to guide the filtering phase, ATLAS can efficiently and effectively explore and derive a set of promising architectures for further training-based evaluation. More details and the pseudocode of the filtering phase are provided in Appendix G.2.

## 3.3 ARCHITECTURE REFINEMENT VIA TRAINING-BASED ARCHITECTURE EVALUATION

Only using ExpressFlow can not accurately indicate the actual performance of a given architecture due to the approximated evaluation. To guarantee high-performing NAS, we further introduce an architecture refinement phase. Specifically, instead of searching the architecture solely based on the ExpressFlow scores, we rank the architectures by their scores in the filtering phase to filter out less promising architectures and only keep the top $K$ architectures with the highest scores. Then, we adopt *training-based architecture evaluation* in the refinement phase to more accurately identify the optimal architecture from the $K$ most promising candidate architectures.

Since fully training each of the $K$ architectures is costly, to expedite the refinement phase, we design a budget-aware algorithm to schedule the training process. In particular, the scheduling algorithm is based on *successive halving* (SUCCHALF) Jamieson & Talwalkar (2016). It begins by allocating an equal, minimal budget for training each architecture. After each round, a fraction (typically half) of the top-performing architectures are kept for further training, and the budgets for the next round of training are doubled. This procedure is repeated until only one single architecture remains. SUCCHALF offers two main benefits. First, it quickly identifies and discards unpromising architectures using only a few training epochs, directing budgets to more promising architectures. Second, it supports parallel training, enhancing efficiency as architectures to be evaluated in each round can be trained in parallel. More details and the pseudocode of the refinement phase are provided in Appendix G.2.

### 3.4 ANYTIME NEURAL ARCHITECTURE SEARCH

In ATLAS, the filtering phase offers more efficient, yet less effective, explorations of a vast set of architectures, denoted as $M$, and the refinement phase enables more effective, yet less efficient, exploitation of a small set of promising architectures, specifically denoted as $K$. To balance these two phases and achieve anytime NAS for tabular data, we propose a novel budget-aware coordinator to jointly optimize the two phases. A strategic objective function is introduced to optimize the performance of the searched architecture. Particularly, we mathematically define constraints to ensure strict adherence to the given time budget $T_{max}$. These constraints collectively determine the time allocation for both phases and the values of $M$, $K$, and $U$.

The two-phase optimization scheme is formalized in Equation 2. For clarity, we denote $t_1$ as the time required to score an architecture using ExpressFlow with a single batch of data, and $t_2$ as the time to train an architecture for a single epoch. $T_1$ and $T_2$ represent the time allocated to the filtering and refinement phases respectively. To sequentially explore $M$ architectures, the filtering phase takes $T_1 = t_1 \cdot M$. As for the refinement phase, the strategy SUCCHALF used only retains the top $1/\eta$ architectures after each training round. Specifically, with $K$ candidate architectures retained, SUCCHALF allocates $K \cdot U \cdot t_2$ time to evaluate these architectures during the initial training round, where each architecture is trained for $U$ epochs. This process iterates for $\lfloor \log_\eta K \rfloor$ training rounds until one single architecture remains, with each round allocated an equal amount of time, i.e., $K \cdot U \cdot t_2$. Hence, the total time for the refinement phase is $T_2 = K \cdot U \cdot t_2 \cdot \lfloor \log_\eta K \rfloor$.

$$
\begin{aligned}
\text{max} \quad & p = \text{ATLAS}(\text{filter}(M, t_1), \text{refinemnet}(K, U, t_2, \eta)) \\
\text{s.t.} \quad & T_1 + T_2 \leq T_{max} \\
\text{where} \quad & T_1 = t_1 \cdot M; K \leq M \\
& T_2 = K \cdot U \cdot t_2 \cdot \lfloor \log_\eta K \rfloor
\end{aligned}
\tag{2}
$$

To strike a balance between the filtering and refinement phases, we assess the sensitivity of $M/K$ and $U$ in relation to the performance of the final searched architecture. Empirical findings demonstrate that setting $M/K \approx 30$ and $U = 2$ yields consistently better search performance, which is detailed in Appendix E. Based on these relationships, the coordinator can thus determine the value of $M$ and $K$ for the filtering phase and refinement phase respectively for any predefined $T_{max}$.

## 4 EXPERIMENTS

### 4.1 NAS-BENCH-TABULAR

#### 4.1.1 SEARCH SPACE STATISTICS

Figure 2 shows the empirical cumulative distribution function (ECDF) of training and validation AUC recorded across all architectures. Architectures have median validation AUC values of 0.9772, 0.8014, and 0.6269 on the Frappe, Criteo, and Diabetes datasets, while the globally optimal architecture yields AUC values of 0.9814, 0.8033, and 0.6750 respectively. These findings are consistent with the performance benchmarks reported in the state-of-the-art study Cai et al. (2021); Wang et al. (2023a), confirming the validation of our training configurations.

Further, we examine the correlation between the parameter count of architecture and their validation AUC across all three datasets. The results are shown in Figure 3. Notably, the architecture's parameter

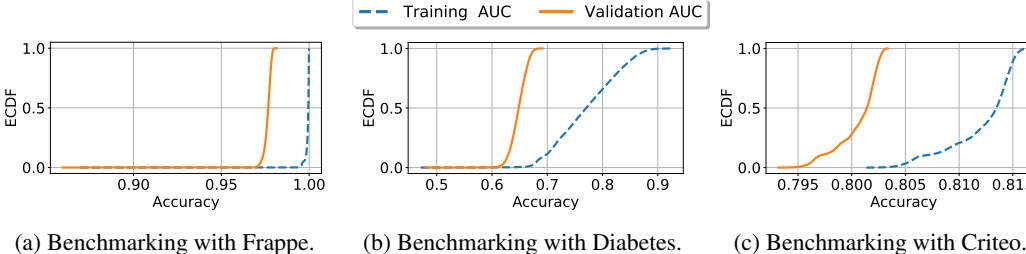

(a) Benchmarking with Frappe.  (b) Benchmarking with Diabetes.  (c) Benchmarking with Criteo.

Figure 2: The empirical cumulative distribution (ECDF) of the training and validation AUC.

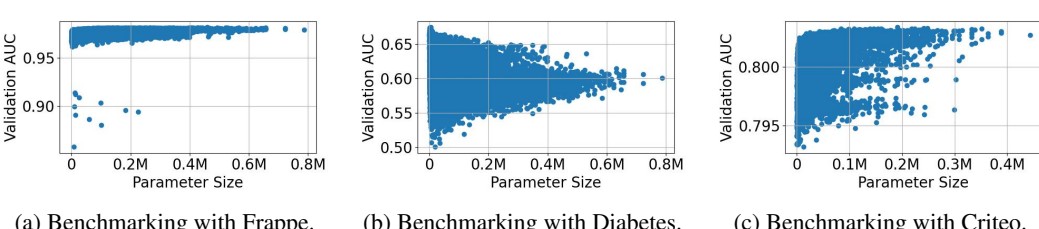

(a) Benchmarking with Frappe.  (b) Benchmarking with Diabetes.  (c) Benchmarking with Criteo.

Figure 3: Validation AUC vs. the number of trainable parameters.

count does not strongly correlate with their validation AUC, highlighting the importance of topology searching (sizes of each hidden layer in DNN) in finding high-performing architectures, and the necessity of NAS for tabular data.

### 4.1.2 BENCHMARKING TRAINING-BASED NAS APPROACHES

In this section, we evaluate four representative searching strategies on NAS-Bench-Tabular, serving as benchmarks for evaluating future NAS algorithms on our datasets. The four searching strategies include Random Search (RS), Regularized Evolution (RE), Reinforcement Learning (RL), and Bayesian Optimization with HyperBand (BOHB). For RE, we set the population size to 10 and the sample size to 3. In the RL setup, we employ a categorical distribution for each hidden layer size and optimize probabilities using policy gradient methods. All these searching strategies explore the same search space and cooperate with the training-based evaluation that queries the validation AUC from NAS-Bench-Tabular directly. With NAS-Bench-Tabular, NAS algorithms can significantly reduce search times to seconds, e.g., each of the four searching strategies can explore over 1k architectures in around 15 seconds, highlighting the benefits of using our NAS-Bench-Tabular. As shown in Figure 4, RE targets the high-performing architecture upon exploring around $10^3$ architectures and it is more efficient as analyzed in Section 3.2. We therefore adopt RE as the search strategy in ATLAS.

### 4.2 TRAINING-FREE ARCHITECTURE EVALUATION METRICS

In this section, we empirically measure the effectiveness and efficiency of nine existing TRAILERs, initially proposed for vision tasks, when applied to DNNs for tabular data. To evaluate the effectiveness, we quantitatively measure the *Spearman Rank Correlation Coefficient (SRCC)* between the scores computed by TRAILERs and the actual performance (e.g., AUC) of candidate architectures across three datasets. The effectiveness of a TRAILER is then defined by its capacity to consistently uphold a high correlation across various datasets.

Table 1 shows that NASWOT, SNIP, SynFlow, and ExpressFlow consistently achieve SRCC of above 0.6 across various datasets, demonstrating their effectiveness in approximating architecture performance. In contrast, other TRAILERs present lower SRCC, suggesting a weaker correlation between their score and the actual architecture performance. ExpressFlow outperforms all other TRAILERs, with an average rank of 1.0. This minimal SRCC variation for ExpressFlow across datasets further confirms its reliable architecture performance characterization and strong transferability. The superior performance of ExpressFlow can be attributed to its capability to characterize both trainability and expressivity, as theoretically analyzed in Section 3.2 and Appendix G.3. Appendix D provides further

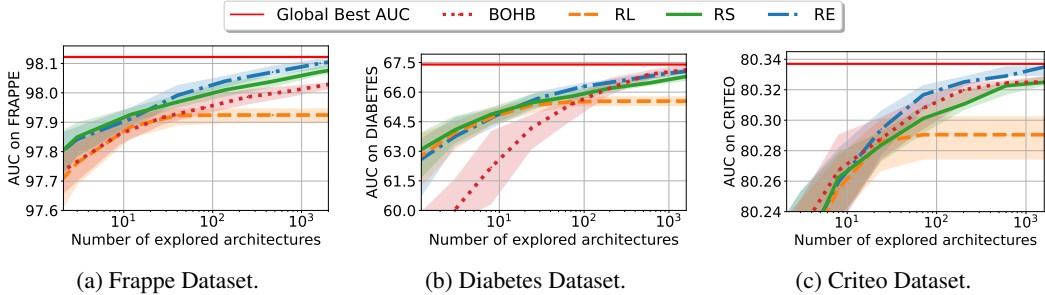

(a) Frappe Dataset.      (b) Diabetes Dataset.      (c) Criteo Dataset.

Figure 4: Benchmarking of the four search strategies using NAS-Bench-Tabular. The x-axis denotes the number of explored architectures, and the y-axis denotes the median value of the best AUC achieved across 100 runs.

Table 1: SRCC of TRAILERs measured on three benchmark datasets.

| | Grad Norm | NAS WOT | NTK Cond | NTK Trace | NTK TrAppx | Fisher | GraSP | SNIP | SynFlow | **ExpressFlow** |
|---|---|---|---|---|---|---|---|---|---|---|
| Frappe | 0.45 | 0.61 | -0.77 | 0.54 | 0.13 | 0.48 | -0.27 | 0.68 | 0.77 | **0.82** |
| Diabetes | 0.39 | 0.63 | -0.56 | 0.37 | 0.31 | 0.21 | -0.23 | 0.62 | 0.68 | **0.73** |
| Criteo | 0.32 | 0.69 | -0.66 | 0.46 | 0.01 | 0.41 | -0.18 | 0.78 | 0.74 | **0.90** |
| Avg Rank | 7.3 | 4.0 | 4.0 | 6.3 | 9.3 | 8.0 | 9.0 | 3.3 | 2.6 | **1.0** |

ablation study on initialization methods, batch size, recalibration weight, etc., and visualization between AUC and the score of TRAILERs. More experiments about TRAILERs computational efficiency, search cost, and transferability to other data types are provided in Appendix G.1.

To further investigate the effectiveness of ExpressFlow, we use the RE search strategy with ExpressFlow as a performance estimator to identify higher-scored architectures, and then record their AUC from NAS-Bench-Tabular. As shown in Figure 5, the search strategy continually explores architectures with higher ExpressFlow scores as more time budget is provided. However, higher-scored architectures do not necessarily promise higher AUC based on the observation that more budgets result in higher-scored but lower-AUC architecture. Thus, solely relying on the filtering phase cannot achieve anytime NAS on tabular data where a larger time budget should ideally lead to better-performing architectures. We, therefore, design a refinement phase and a budget-aware coordinator to jointly optimize two phases towards anytime NAS as analyzed in Sections 3.3 and 3.4.

### 4.3 ANYTIME NAS ON TABULAR DATA

Last, we benchmark our approach ATLAS incorporating the filtering phase, the refinement phase, and the budget-aware coordinator, against the training-based NAS utilizing RE as the search strategy (referred to as RE-NAS) and TabNAS Yang et al. (2022).

To evaluate the anytime performance of different approaches, we adjust the $T_{max}$ to span from seconds to hours, to investigate three primary questions: (1) How much time is required by each approach to search for an architecture near the global best AUC? (2) Can the NAS approach be completed within any $T_{max}$? (3) Does the performance of the searched architecture maintain stability or exhibit improvement with an increased time budget?

The results are shown in Figure 6. First, regarding time usage in searching for the global best AUC, ATLAS outperforms RE-NAS by speedup of 82.75x, 1.75x, and 69.44x across the Frappe, Diabetes, and Criteo datasets, achieving AUCs of 0.9814, 0.6750, and 0.8033, respectively. Both RE-NAS and TabNAS employ training-based architecture evaluation, slowing the exploration and identification of the optimal-performing architecture. While ATLAS employs both efficient training-free and effective training-based evaluations with a joint optimization, it can rapidly explore a vast of architectures, filter out less-promising ones, and allocate more budgets to exploit high-potential architectures. Consequently, ATLAS requires less time to search for the global best AUC.

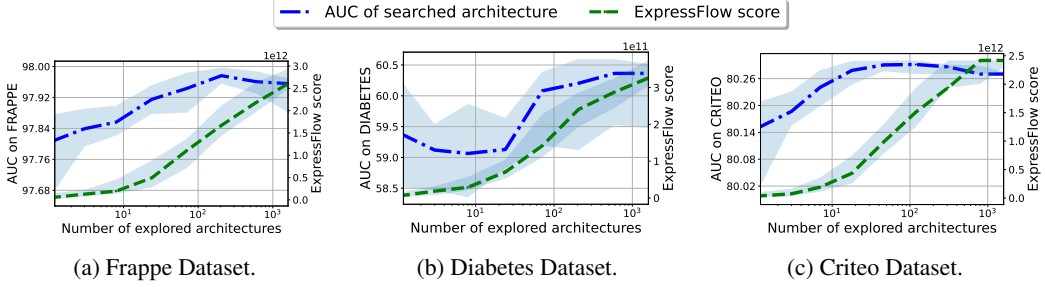

Figure 5: The relationship between the ExpressFlow score and the AUC of searched architecture.

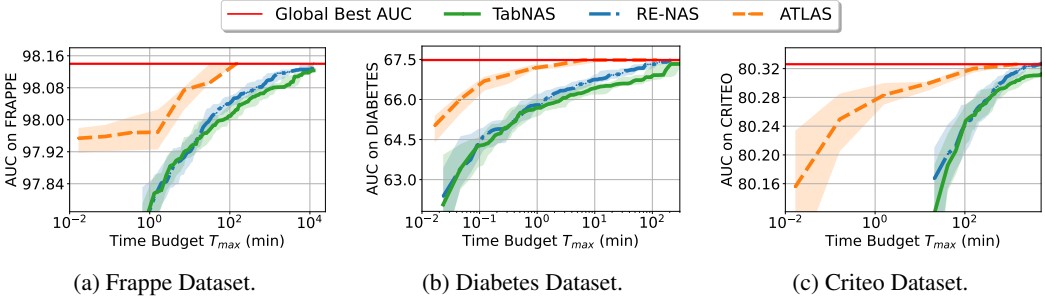

Figure 6: Anytime performance of ATLAS compared with RE-NAS and TabNAS.

Second, in terms of completing within any $T_{max}$, RE-NAS and TabNAS generally require 5 to 10 minutes to evaluate a single architecture, which violates the anytime NAS requirement when the $T_{max}$ is small, e.g., less than 5 minutes. ATLAS shows strong anytime performance by consistently completing NAS even when the $T_{max}$ is only a few seconds. This is attributed to the budget-aware coordinator (Section 3.4), which decides the allocation of $T_{max}$ between the filtering and refinement phases, with the mathematical constraints ensuring the total time usage is under $T_{max}$.

Last, with regard to performance consistency for a larger time budget, all three NAS approaches consistently identify equal or superior-performing architectures with incrementally larger $T_{max}$. ATLAS demonstrates consistently superior performances, which discover higher-performing architectures across all $T_{max}$ compared to RE-NAS and TabNAS. This is attributed to the design of our strategic objective function as introduced in Section 3.4, which balances efficient exploration in the filtering phase and effective exploitation in the refinement phase. Specifically, in the filtering phase, ExpressFlow efficiently facilitates the evaluation of a larger number of architectures within the same $T_{max}$, thereby enabling a more extensive exploration of architectures towards the global best AUC. Further, the refinement phase provides a comparatively precise performance assessment of each architecture, thereby mitigating the inherent uncertainty in the filtering phase.

We further compare ATLAS with more baselines including various combinations of training-free and training-based methods and one-shot NAS in Appendix F.

## 5 CONCLUSION

In this work, we introduce an anytime NAS approach ATLAS tailored for tabular data. ATLAS equips machine learning practitioners with the capability of ascertaining high-performance architectures within any given time budget and further refining these architectures as larger time budgets are given. We first design a comprehensive search space, denoted as NAS-Bench-Tabular, to serve as a benchmarking platform for diverse NAS algorithms. Then, we conduct a comprehensive evaluation of several TRAILERs and propose a novel metric, ExpressFlow, which characterizes both the trainability and expressivity of an architecture. Based on these foundations, we present ATLAS, which leverages the advantages of efficient training-free and effective training-based architecture evaluation through a novel filtering-and-refinement optimization scheme with joint optimization. Empirical results show that ATLAS significantly accelerates the search for the globally optimal architecture and achieves anytime NAS on tabular datasets.

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

# CONTENTS OF THE APPENDICES

**Appendix A.** Literature review of NAS, DNN on tabular data, and architecture properties.

**Appendix B.** NAS research best practices checklist Lindauer & Hutter (2019).

**Appendix C.** Dataset statistics, hyperparameter configurations, and search space backbone for building NAS-Bench-Tabular.

**Appendix D.** Ablation studies and visualization of ExpressFlow.

**Appendix E.** Empirically analysis of the budget-aware coordinator's effectiveness and necessity.

**Appendix F.** Comparison of ATLAS with additional baselines, encompassing various combinations of training-free and training-based methods, as well as one-shot NAS.

**Appendix G.1.** A comprehensive comparison of different TRAILERs.

**Appendix G.2.** Pseudocode for the filtering and refinement phases in our approach.

**Appendix G.3.** Theoretical analysis of the trainability and expressivity of ExpressFlow.

**Appendix G.4.** Data-agnostic property and efficient computational pattern of ExpressFlow.

**Appendix H.** Summary of Notation and Terminology.

**Appendix I.** Links to source code and datasets.

# A    LITERATURE REVIEW

## A.1    NEURAL ARCHITECTURE SEARCH (NAS)

Neural Architecture Search (NAS) is designed to automate the discovery of architectures optimized for specified datasets, thereby eliminating the need for manual design and experimentation. The process often entails the search for connection patterns within a pre-defined architectural backbone, such as Fully-connected neural network Yang et al. (2022) or cell-based convolutional neural network Ying et al. (2019); Dong & Yang (2020).

A critical component of NAS is the architecture evaluation. Initial research in NAS Zoph & Le (2017); Baker et al. (2017) primarily relied on the time-consuming and resource-intensive process of fully training each architecture to convergence. Several methods have been proposed to address this issue and to mitigate costs Lee et al. (2023); Shala et al. (2023); Li et al. (2023); Bohdal et al. (2023). They generally fall into three categories: NAS with performance prediction, NAS based on weight sharing, and NAS with training-free architecture evaluation.

Performance prediction in NAS involves training a model to forecast the final performance of an architecture based on features derived from hyperparameters, architectural structures, and partially trained architectures. This category of methods shows improvements in searching efficiency but it is limited in tuning the predictors and is hard to enhance generalizability Siems et al. (2020); White et al. (2021).

In contrast, NAS based on weight sharing seeks to identify a subgraph within a larger computation graph Pham et al. (2018). This allows multiple sampled subgraphs sharing the same computation unit to utilize a common set of weights. However, the individual sampling and training procedure of each discrete subgraph leads to an increased number of architectures to be trained. And inheriting weights from the larger computational graph does not necessarily ensure improved training efficiency Chu et al. (2021). To further reduce the number of architectures requiring training, Liu et al. (2019b) proposed shifting the search from a discrete space to a continuous one. This enables gradient optimization to expedite the search process.

Last, training-free architecture evaluation estimates architecture performance by calculating certain statistics of the architecture at initialization without requiring full training Li et al. (2023); White et al. (2023; 2021); Abdelfattah et al. (2021). One of the main advantages of the training-free evaluation is its extremely high computational efficiency, requiring only a single forward and/or backward computation.

The ATLAS approach proposed in this work effectively combines the strengths of both paradigms of training-free and training-based architecture evaluation to provide high efficiency and effective architecture searching.

## A.2    DNN ON TABULAR DATA

Different approaches have attempted to apply DNN techniques to tabular data, ranging from DNN design Levin et al. (2023); Cai et al. (2021); Luo et al. (2021) to Automated Machine Learning (AutoML) on tabular data Fusi et al. (2018); Olson & Moore (2016); Yang et al. (2019). However, NAS on tabular data has been relatively less explored. Recently, AgEBO-Tabular Égelé et al. (2021) and TabNAS Yang et al. (2022) investigated the application of NAS on tabular data to achieve more efficient and higher-performing architectures. Specifically, AgEBO-Tabular integrates NAS with aging evolution in a search space that includes multiple branches and hyperparameter tuning using Bayesian optimization. In comparison, TabNAS aims to identify high-performing architectures from Fully-connected neural network under specified resource constraints by utilizing reinforcement learning with reject sampling. It shows that simple multiple layers Fully-connected neural network can already yield outstanding performance. However, both approaches rely on training-based architecture evaluation, and cannot achieve anytime NAS on tabular data.

ATLAS in this paper introduces a novel filtering-and-refinement optimization scheme with joint optimization of both training-free and training-based architecture evaluation and can achieve anytime NAS on three tabular datasets. To the best of our knowledge, this is the first work in this direction.

## A.3 ARCHITECTURE PROPERTIES

Architecture performance is influenced by two factors: trainability and expressivity. Trainability measures the extent to which gradient descent can effectively optimize the architecture. Expressivity denotes the complexity of the function that the architecture can model. Many training-free evaluation approaches estimate the architecture performance by characterizing both properties.

More recently, Chen et al. (2021) proposes to quantify the expressivity of a ReLU-based DNN by computing the number of linear regions that the architecture can divide for a batch of data. Likewise, NASWOT Mellor et al. (2021) characterizes expressivity by measuring the distance between the vectors of activation patterns for any two samples within a batch. A greater distance suggests a higher capability to distinguish different samples, indicating good expressivity.

The application of the Neural Tangent Kernel (NTK) as a measure of trainability has recently been explored, given a batch of data $X_B$, NTK Jacot et al. (2018); Arora et al. (2019); Allen-Zhu et al. (2019) characterizes the complexities of the training dynamics at initialization, which is defined as $\Theta(X_B, X_B; \boldsymbol{\theta}) = \nabla_{\boldsymbol{\theta}} f(X_B; \boldsymbol{\theta}) \nabla_{\boldsymbol{\theta}} f(X_B; \boldsymbol{\theta})^T$. NTK-related metrics are adopted in many training-free evaluation approaches, such as NTKTrace Shu et al. (2022b), NTKTraceAppx Shu et al. (2022a), and NTKCond Chen et al. (2021).

The trainability has also been studied in the context of network pruning Tanaka et al. (2020); Wang et al. (2020a); Lee et al. (2019), which identifies and prunes less significant parameters. The notion of synaptic saliency Tanaka et al. (2020) is proposed to quantify each parameter's importance, defined as $\Phi(\boldsymbol{\theta}) = f(\frac{\partial \mathcal{L}}{\partial \boldsymbol{\theta}}) \odot g(\boldsymbol{\theta})$. Different metrics basically differ in $f(\cdot)$ and $g(\cdot)$, e.g., $\Phi(\boldsymbol{\theta}) = |\frac{\partial \mathcal{L}}{\partial \boldsymbol{\theta}}| \odot |\boldsymbol{\theta}|$ in SNIP Lee et al. (2019), $\Phi(\boldsymbol{\theta}) = -(H \frac{\mathcal{L}}{\partial \boldsymbol{\theta}}) \odot \boldsymbol{\theta}$ in GraSP Wang et al. (2020a), and $\Phi(\boldsymbol{\theta}) = \frac{\partial \mathcal{L}}{\partial \boldsymbol{\theta}} \odot \boldsymbol{\theta}$ in SynFlow Tanaka et al. (2020), where $H$ is the Hessian vector. Similarly, Fisher Turner et al. (2020) quantifies the performance by aggregating layer Fisher information Theis et al. (2018).

In our study, we introduce a novel, training-free architecture evaluation metric called ExpressFlow to capture both trainability and expressivity. It demonstrates higher correlations across three tabular datasets.


## C  NAS-BENCH-TABULAR

### C.1  DATASETS

We use three datasets as shown in Table 2, and briefly summarize them as follows:

**Frappe** [1] is a dataset from the real-world application recommendation scenario, which incorporates context-aware app usage logs consisting of 96,203 tuples from 957 users across 4,082 apps used in various contexts. For each positive app usage log, Frappe generates two negative tuples, resulting in a total of 288,609 tuples. The learning objective is to predict app usage based on the context, encompassing 10 semantic attribute fields with 5,382 distinct numerical and categorical embedding vectors.

**UCI Diabetes** [2] (Diabetes) encompasses a decade (1999-2008) of clinical diabetes encounters from 130 US hospitals. This dataset aims to analyze historical diabetes care to enhance patient safety and deliver personalized healthcare. With 101,766 encounters from diabetes-diagnosed patients, the primary learning objective is to predict inpatient readmissions. This dataset consists of 43 attributes and 369 distinct numerical and categorical embedding vectors, including patient demographics and illness severity factors like gender, age, race, discharge disposition, and primary diagnosis.

**Criteo** [3] is a CTR benchmark consisting of attribute values and click feedback for millions of display advertising. The learning objective is to predict if a user will click a specific ad in the context of a webpage. This dataset has 45,840,617 tuples across 39 attribute fields with 2,086,936 distinct numerical and categorical embedding vectors. These include 13 numerical attribute fields and 26 categorical attributes.

Table 2: Dataset Statistics

| Dataset | # Class | # Sample | # Feature | Task |
|---------|---------|----------|-----------|------|
| Frappe | 2 | 288,609 | 10 | App Recommendation |
| Diabetes | 2 | 101,766 | 43 | Healthcare Analytics |
| Criteo | 2 | 45,840,617 | 39 | CTR Prediction |

### C.2  TRAINING HYPERPARAMETERS

Table 3: Training Hyperparameters

| Dataset | batch size | learning rate | learning rate schedule | optimizer | training epoch | iteration per epoch | loss function |
|---------|-----------|---------------|------------------------|-----------|----------------|---------------------|---------------|
| Frappe | 512 | 0.001 | cosine decay | Adam | 20 | 200 | BCELoss |
| Diabetes | 1024 | 0.001 | cosine decay | Adam | 1 | 200 | BCELoss |
| Criteo | 1024 | 0.001 | cosine decay | Adam | 10 | 2000 | BCELoss |

Training architectures on each dataset necessitates the optimal configuration of hyperparameters such as training epochs, iterations, learning rate, etc. For each dataset, we employ the grid search to fine-tune the training hyperparameters. Our goal is to ensure that the three typical architectures within our search space, i.e., from the smallest to medium and to the largest sizes, achieve DNN performances that are consistent with the results reported in Cai et al. (2021); Kadra et al. (2021); Klambauer et al. (2017); Fernández-Delgado et al. (2014). We configure the hyperparameters as detailed in Table 3. For the Adam optimizer, we adopt the settings $\beta_1 = 0.9$, $\beta_2 = 0.999$, decay = 0, and $\epsilon =$ 1e-8, as proposed in Yang et al. (2022); Kingma & Ba (2014).

---

[1] https://www.baltrunas.info/research-menu/frappe

[2] https://archive.ics.uci.edu/ml/datasets

[3] https://labs.criteo.com/2014/02/kaggle-display-advertising-challenge-dataset/

## C.3 SEARCH SPACE DESIGN

We fix the number of layer $L$ in DNN to be four for all three tabular datasets and set the candidate set of layer sizes $\mathcal{H}$ as follows:

- Frappe and Diabetes: $\mathcal{H} = [8, 16, 24, 32, 48, 64, 80, 96, 112, 128, 144, 160, 176, 192, 208, 224, 240, 256, 384, 512]$.

- Criteo: $\mathcal{H} = [8, 16, 32, 48, 64, 112, 144, 176, 240, 384]$.

The size of the search space for Criteo is relatively smaller, with only $10^4$=10,000 architectures, but the best-performing architecture found already achieves the state-of-art performance compared with related works Cai et al. (2021); Kadra et al. (2021); Levin et al. (2023); Fernández-Delgado et al. (2014).

## D  ADDITIONAL EXPERIMENTS OF EXPRESSFLOW

In this section, we present detailed experiments covering an ablation study and correlation visualization. All experiments are conducted using PyTorch on a local server equipped with an Intel(R) Xeon(R) Silver 4214R CPU (12 cores), 128 GB of memory, and 8 GeForce RTX 3090 GPUs.

### D.1  ABLATION STUDIES AND ANALYSIS

**Impacts of Parameter Positivity.** As neuron saliency is calculated at the architecture initialization as illustrated in Section 3.2, we examine the impact of the parameter's sign on the effectiveness of ExpressFlow. Specifically, we score each architecture with ExpressFlow using either the parameter's absolute value or its default. We then compare the correlations of both approaches with the architecture's AUC. For consistency across multiple datasets, we maintain a fixed batch size $B$ of 32 and employ the Xavier initialization method. Interestingly, the results in Table 4 reveal that parameter positivity significantly impacts ExpressFlow's effectiveness, leading to a correlation increase of up to 78.8% in the Criteo dataset. Therefore, before computing the ExpressFlow score, we set all architecture parameters to their absolute values.

Table 4: Impacts of Parameter Positivity

| Dataset | Frappe | Diabetes | Criteo |
|---|---|---|---|
| ExpressFlow with positive $\mathbf{w}$ | 0.8364 | 0.7124 | 0.8978 |
| ExpressFlow with default $\mathbf{w}$ | 0.5175 | 0.5901 | 0.5020 |

**Impacts of Initialization Method.** Based on the observed advantages of using absolute parameter values, we further explore how various initialization methods influence the ExpressFlow correlation. With the batch size of $B$=32, we examine three different initializations: LeCun LeCun et al. (2002), Xavier Glorot & Bengio (2010), and He He et al. (2015). Table 5 shows that while all three initializations produce comparable correlation values, the He initialization consistently outperforms the others on the Criteo and Diabetes datasets. This is because He is particularly designed for ReLU activation functions, and ReLU is exactly what our search space backbone uses as detailed in Section 3.1. Therefore, we adopt He initialization as the default setting in our approach.

Table 5: Impacts of Initialization method

| Dataset | LeCun LeCun et al. (2002) | Xavier Glorot & Bengio (2010) | He He et al. (2015) |
|---|---|---|---|
| Frappe | 0.8175 | 0.8364 | 0.8150 |
| Diabetes | 0.7335 | 0.7124 | 0.7336 |
| Criteo | 0.8823 | 0.8978 | 0.9005 |

**Impacts of Batch Size** $B$. Considering that ExpressFlow depends on a forward and a backward computation on a mini-batch of data, we extend our analysis to measure the effectiveness of ExpressFlow with respect to different batch sizes. Experiments are conducted on three datasets, with batch sizes varying from 4 to 128. For each setting, we fix the initialization method to He and repeat the experiment five times, and then we record the median correlation value. The results in Table 6 indicate that the batch size has minimal influence on the final correlation of ExpressFlow, suggesting that ExpressFlow can potentially be computed efficiently by using only small data batches with $B$=4.

Table 6: Impacts of Batch Size

| Dataset | $B = 4$ | $B = 8$ | $B = 16$ | $B = 32$ | $B = 64$ | $B = 128$ |
|---------|---------|---------|----------|----------|----------|-----------|
| Frappe | 0.8154 | 0.8152 | 0.8150 | 0.8150 | 0.8150 | 0.8149 |
| Diabetes | 0.7335 | 0.7336 | 0.7335 | 0.7336 | 0.7336 | 0.7336 |
| Criteo | 0.8990 | 0.8998 | 0.9008 | 0.9005 | 0.9009 | 0.9009 |

**Data-Agnostics.** Given the observation that the correlation of ExpressFlow exhibits limited sensitivity to different batch sizes, we set the batch size $B$=4 and shift our focus to explore the influence of data samples on the ExpressFlow correlation. Instead of employing a random data sample of dimension $d$ as the input, we utilize an all-one vector with the same dimension, denoted as $\mathbf{1}_d$, as the input data sample and measure the ExpressFlow correlation. The results presented in Table 7 suggest that utilizing an all-one vector yields results remarkably similar to those obtained through real data samples shown in Table 6. This evidence supports the assertion that ExpressFlow captures the potentially well-performing architecture in a data-agnostic manner, which confirms its good transferability across different datasets.

Table 7: Data Agnostics Property of ExpressFlow

| Dataset | Frappe | Diabetes | Criteo |
|---------|--------|----------|--------|
| Data sample $\mathbf{1}_d$ | 0.8151 | 0.7336 | 0.8991 |

**Impact of Recalibration Weight of Neuron Saliency** Since we perform a weighted aggregation of neuron saliency to compute the ExpressFlow score, i.e., each neuron saliency at $l$-th layer with $\frac{\mathcal{K}_l}{\ell(z^l)}$ as shown in Section 4.2. We further examine the impacts of varying recalibration weight on characterizing the performance of the architecture on three datasets. Specifically, we contrast ExpressFlow, which considers both width and depth in the recalibration weight, with two variations: (1) ExpressFlow leveraging only the width, labeled as recalibrated by $\mathcal{K}$; (2) ExpressFlow focusing solely on depth, referred to as recalibrated by $\frac{1}{\ell(z^l)}$; Table 8 reveals that ExpressFlow, when considering both width and depth in recalibration weight, consistently outperforms its other variations in terms of correlation across all datasets.

Table 8: Impacts of Recalibration Weights of Neuron Saliency

| Dataset | recalibrated by $\mathcal{K}$ | recalibrated by $\frac{1}{\ell(z^l)}$ | recalibrated by $\frac{\mathcal{K}_l}{\ell(z^l)}$ |
|---------|-------------------------------|----------------------------------------|----------------------------------------------------|
| Frappe | 0.7007 | 0.7296 | 0.8151 |
| Diabetes | 0.6772 | 0.6900 | 0.7336 |
| Criteo | 0.6402 | 0.6414 | 0.8991 |

## D.2 VISUALIZATION OF CORRELATION FOR TRAILERs

For each of TRAILER, we randomly sample 4000 architectures from the NAS-Bench-Tabular. We plot their correlations between validation AUC after training and the TRAILER score computed at the architecture initialization. The results are shown in Figures 7 to 16.

Our findings demonstrate that scores of GradNorm 7, NASWOT 8, NTKTrace 10, NTKTraceAppx 11, Fisher 12, SNIP 14, SynFlow 15, and ExpressFlow 16 have a positive correlation with validation AUC across all datasets. This relationship is particularly pronounced for ExpressFlow, which confirms the effectiveness of ExpressFlow for tabular data.

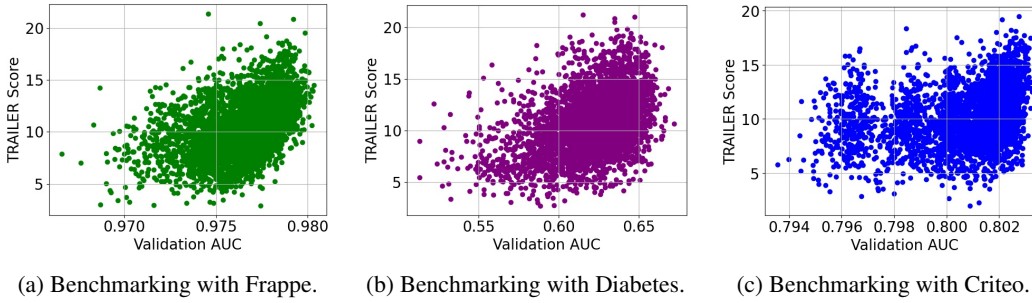

(a) Benchmarking with Frappe.  (b) Benchmarking with Diabetes.  (c) Benchmarking with Criteo.

Figure 7: Score of GradNorm vs. Validation AUC.

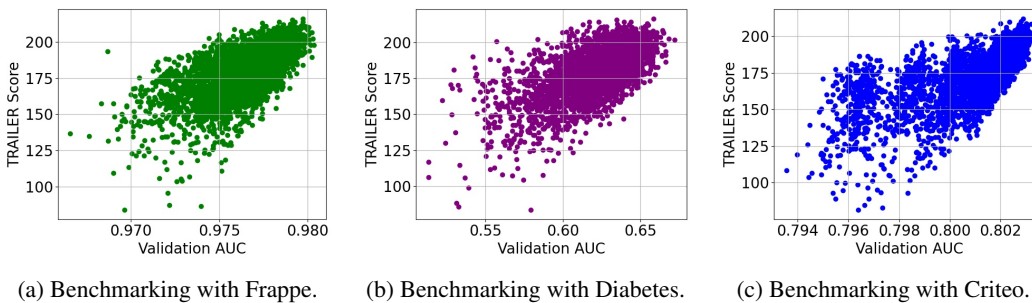

(a) Benchmarking with Frappe.  (b) Benchmarking with Diabetes.  (c) Benchmarking with Criteo.

Figure 8: Score of NASWOT vs. Validation AUC.

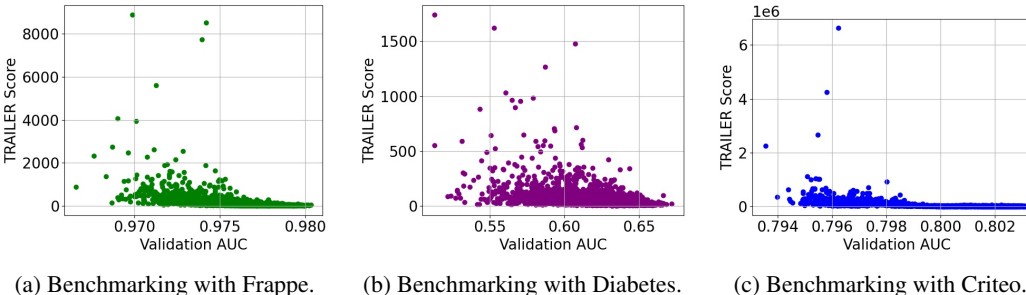

(a) Benchmarking with Frappe.  (b) Benchmarking with Diabetes.  (c) Benchmarking with Criteo.

Figure 9: Score of NTKCond vs. Validation AUC.

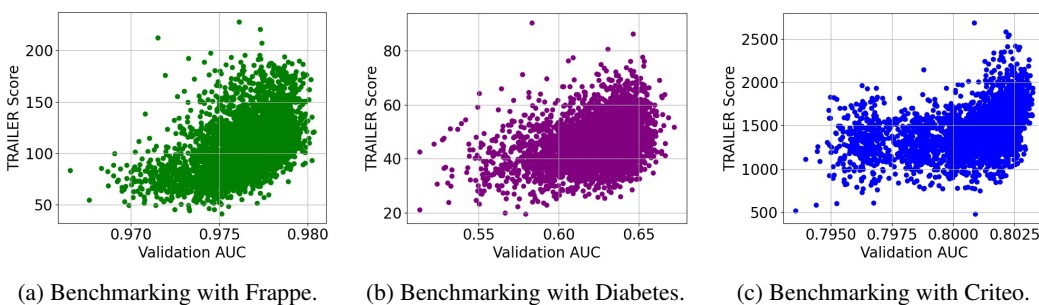

| (a) Benchmarking with Frappe. | (b) Benchmarking with Diabetes. | (c) Benchmarking with Criteo. |

Figure 10: Score of NTKTrace vs. Validation AUC.

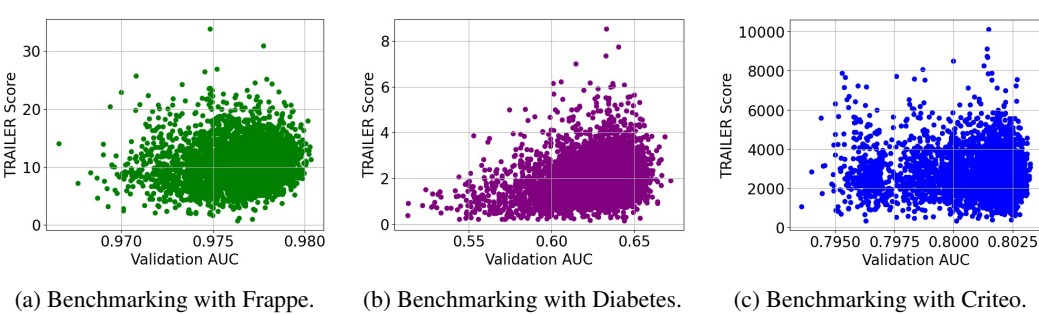

| (a) Benchmarking with Frappe. | (b) Benchmarking with Diabetes. | (c) Benchmarking with Criteo. |

Figure 11: Score of NTKTraceAppx vs. Validation AUC.

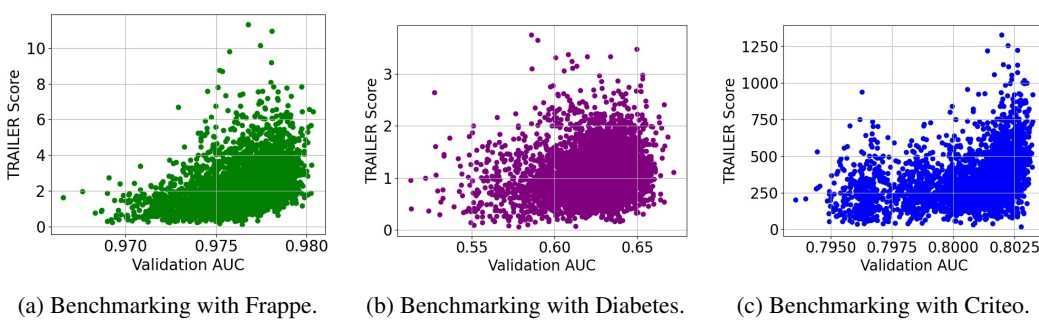

| (a) Benchmarking with Frappe. | (b) Benchmarking with Diabetes. | (c) Benchmarking with Criteo. |

Figure 12: Score of Fisher vs. Validation AUC.

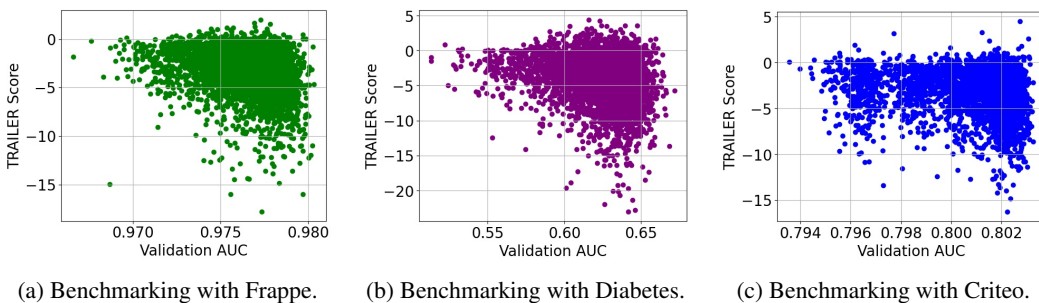

| (a) Benchmarking with Frappe. | (b) Benchmarking with Diabetes. | (c) Benchmarking with Criteo. |

Figure 13: Score of GraSP vs. Validation AUC.

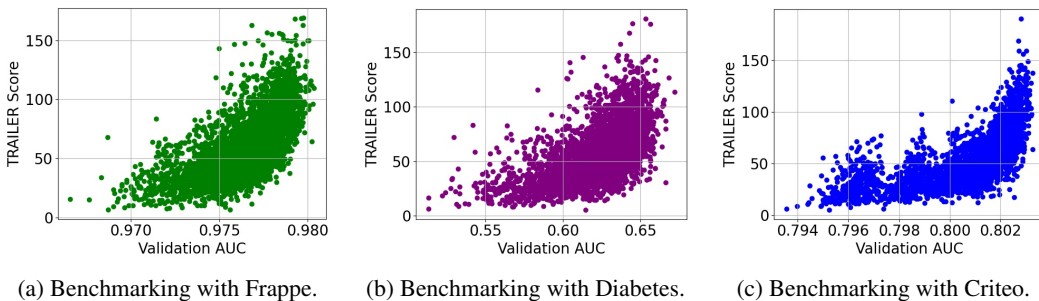

(a) Benchmarking with Frappe.  (b) Benchmarking with Diabetes.  (c) Benchmarking with Criteo.

Figure 14: Score of SNIP vs. Validation AUC.

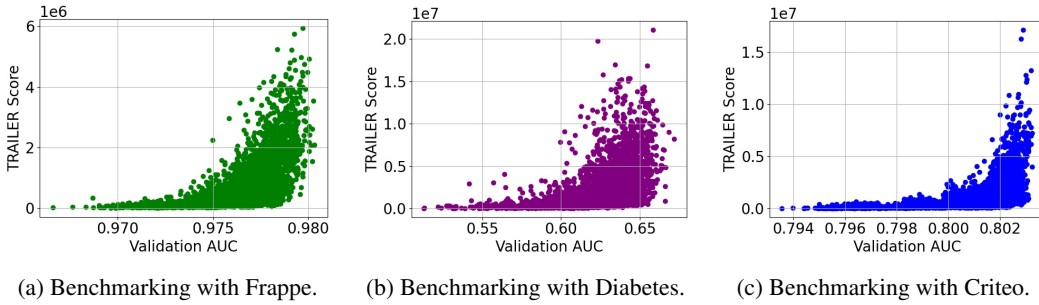

(a) Benchmarking with Frappe.  (b) Benchmarking with Diabetes.  (c) Benchmarking with Criteo.

Figure 15: Score of SynFlow vs. Validation AUC.

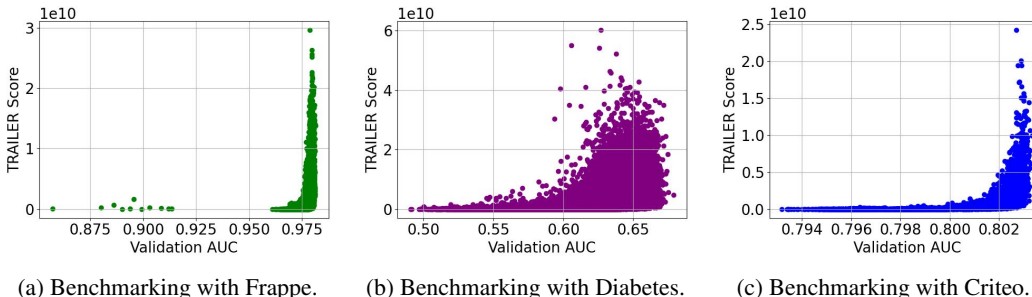

(a) Benchmarking with Frappe.  (b) Benchmarking with Diabetes.  (c) Benchmarking with Criteo.

Figure 16: Score of ExpressFlow vs. Validation AUC.

# E  COORDINATION SCHEME DESIGN AND ANALYSIS

In this section, we present detailed experiments analyzing sensitivity with respect to $M$, $K$, and $U$, and extend our comparisons to include baselines such as zero-cost NAS Abdelfattah et al. (2021) and one-shot NAS Pham et al. (2018). All experiments are conducted on a local server with an Intel(R) Xeon(R) Silver 4214R CPU (12 cores), 128 GB of memory, and 8 GeForce RTX 3090 GPUs.

## E.1  SENSITIVITY OF $M/K$ AND $U$

As explained in Section 3.4, we propose a filtering-and-refinement optimization scheme with joint optimization, combining the strengths of both training-based and training-free paradigms in a more principled way, as well as introduce an objective function defined in equation 2.

The primary challenge lies in striking a balance between $M$ and $K$ given $T_{max}$, which is the number of candidate architectures explored by ExpressFlow in the filtering phase and the number of promising architectures evaluated by training in the refinement phase, respectively. As empirically evaluated in Section 4.2 and  4.3, exploring numerous architectures while neglecting the refinement phase (e.g., $K = 1$) is highly efficient but may result in a sub-optimal architecture being selected since the ExpressFlow in the filtering phase may not accurately indicate the actual performance of a given architecture. In contrast, evaluating each explored architecture for $U$ epochs in training-based methods (e.g., $K = M$) requires training many architectures, which would violate the anytime NAS requirement.

Further, with a fixed time budget for the refinement phase, it is challenging to decide the trade-off between $K$ and $U$, which corresponds to exploiting more architectures with each evaluating fewer epochs, and exploiting fewer architectures with each evaluating more accurately.

To address these challenges, we first examine the trade-off between $K$ and $U$. Specifically, we explore a set of architectures during the filtering phase and then vary the combinations of $K$ and $U$ to measure the achieved AUC and the total training epochs in the refinement phase on two datasets. Figure 17 shows that evaluating each architecture with more epochs can accurately identify the performance. However, this comes at the expense of exploiting fewer architectures, thereby reducing the opportunity to search for higher-performing architectures—such as 98.01% on Frappe and 80.31% on Criteo datasets. Conversely, exploiting more architectures by training each for fewer epochs (e.g., $U$=2) can not only facilitate the search for higher AUC architectures but also decrease the total training epochs. Based on these findings, we set $U$=2 in our ATLAS.

Next, we delve into the trade-offs between $M$ and $K$. With varying $T_{max}$, we examine how the ratio of $M/K$ impacts the AUC of the searched architecture. As shown in Figure 18, $M/K \approx 30$ could yield a good-performing final architecture across different $T_{max}$.

## E.2  NECESSITY AND EFFECTIVENESS OF COORDINATOR

To further illustrate the necessity and effectiveness of the coordinator, we conduct an experiment that sets a fixed search time budget of 100 minutes. We then compare the AUC of architectures searched using various manually set $K$ values with those defined by the coordinator. The results are presented in Table 9. From the table, we can observe that $K$ heavily influences the search performance, and a moderate $K$=309 automatically determined by the coordinator achieves the highest AUC. This confirms that the budget-aware coordinator is effective in jointly optimizing both filtering and refinement phases as well as necessary in achieving anytime NAS on tabular data.

Table 9: Impact of $K$ on the searched AUC. Search for 100 mins.

|  | K=1 | K=10 | K=100 | K=500 | Coordinator (K=309) |
|---|---|---|---|---|---|
| AUC | 0.9789 | 0.9801 | 0.9802 | 0.9791 | **0.9805** |

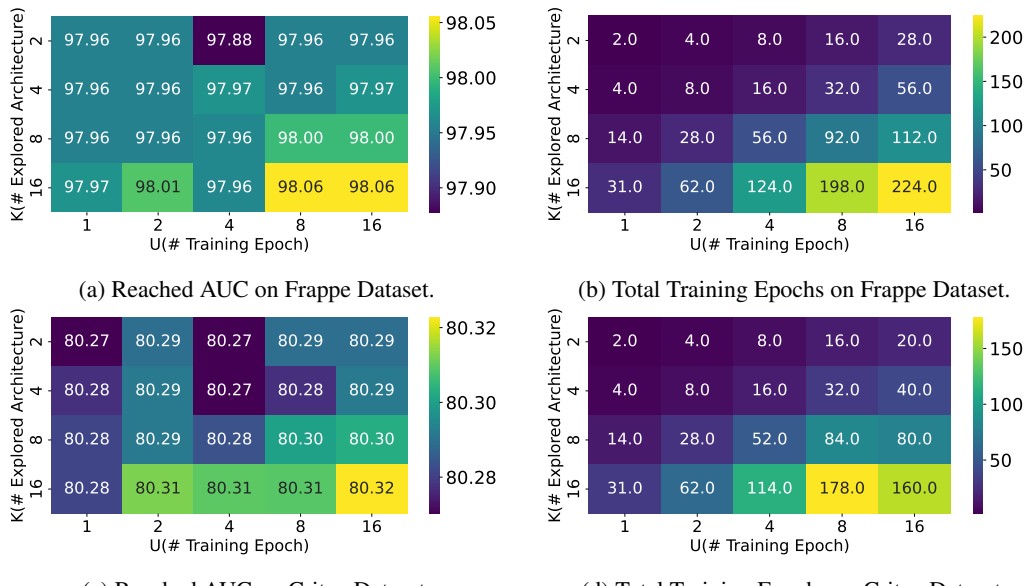

Figure 17: Trade-offs between $K$ and $U$. The ATLAS approach can search for a good-performing architecture with smaller $U$, e.g., $U$=2, across two tabular dataset.

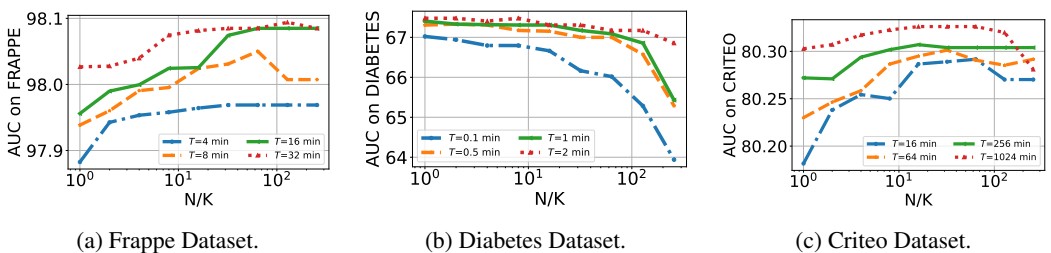

(a) Frappe Dataset.  (b) Diabetes Dataset.  (c) Criteo Dataset.

Figure 18: Trade-offs between $M$ and $K$. The approach can search for a good-performing architecture with $M \approx 30 * K$ across $T_{max}$.

### E.2.1 NOISE IMPACT ON REFINEMENT PHASE

Given that ExpressFlow scores may not provide exact estimations of architecture performance, we then investigate the potential benefits of introducing some random noise during the selection of the top-K architectures for the refinement phase. To examine this, we fix the total search time to 20 minutes and measure the AUC achieved across various *Random Noise Degrees* (RND).

Specifically, given the time budget, the coordinator determines the trade-off between the filtering and refinement phases, which explores $M$ architectures in the train-free filtering phase and keeps the top $K$ for the training-based refinement phase. For each RND, we keep the top $(1 - \text{RND}) \times K$ architectures from the top-K architectures and randomly select another $\text{RND} \times K$ from the search space. These $K$ architectures are then evaluated in the refinement phase.

Results in Table 10 show that a lower degree of noise improves the AUC of the search architecture, which suggests that introducing noise is not beneficial when selecting the top-K architectures, and further confirms the effectiveness of the coordinator.

Table 10: Impact of noise degrees on the searched AUC. Search for 20 mins.

|     | RND = 100% | RND = 70% | RND = 50% | RND = 30% | RND = 0% |
|-----|------------|-----------|-----------|-----------|----------|
| AUC | 0.9792     | 0.9794    | 0.9796    | 0.9799    | **0.9980**2 |

# F COMPARISON WITH ADDITIONAL BASELINES

## F.1 COMPARISON WITH DIFFERENT COMBINATIONS OF TRAINING-FREE AND TRAINING-BASED METHODS

In this section, we compare our approach with two popular NAS approaches that combine both training-free and training-based evaluations. The first is the fully decoupled *warmup* strategy Abdelfattah et al. (2021), which employs training evaluations as proxies to pre-train a search algorithm for subsequent conventional training-based search. The second is the coupled *move proposal* strategy Abdelfattah et al. (2021), which iteratively uses training-free evaluation to quickly propose the next candidate architectures for full training. In contrast, our approach is fundamentally different from theirs. Specifically, in our two-stage optimization, the filtering phase uses training-free metrics and is guided by the RE algorithm, and the refinement phase uses training-based evaluation and is scheduled by a successive-halving algorithm. Our two phases are fully decoupled and are time budget-aware. They are jointly optimized via a coordinator (as detailed in Section 3.4) to support anytime NAS.

To empirically demonstrate the efficiency of our approach, we conduct comparisons on different combination strategies in terms of the time usage for searching for an architecture with a specific AUC. More detailed ablations concerning the combination of different proxies and/or (re)training strategies are also provided. Specifically, we compare ATLAS with training-based (using the search strategy RS, RL, and RE), training-free (using best-performing TRAILERs), Warmup and Move-Proposal (both using RE as the search strategy as in ATLAS, and utilizing best-performing TRAILERs).

The results in Table 11 show that (1) Training-based only and Move-Proposal are the two most time-consuming approaches, as both require iterative and costly full training of architectures; (2) Training-free only approach could not reach the target AUC of 0.9798; (3) our ATLAS achieves the target AUC using only 62 seconds, obtaining a speedup of up to 3.66x and 111.94x compared to Warmup and Move-Proposal; (4) the novel two-phase scheme with a coordinator approach indeed achieves much better search performance than other approaches, outperforming the second-best approach (Warmup+ExpressFlow) by a factor of 3.66x. These findings confirm the necessity of the combination of training-free and training-based NAS, the efficiency of our approach ATLAS, and the effectiveness of our novel decoupled two-phase scheme with a joint optimization approach.

Table 11: Target AUC = 0.9798. N/R: the target AUC is not reached.

| NAS Approaches | Search Cost for AUC 0.9798 (Sec) |
|---|---|
| Training-based (RS) | 21560 |
| Training-based (RL) | N/R |
| Training-based (RE) | 8462 |
| Training-free (SNIP) | N/R |
| Training-free (NASWOT) | N/R |
| Training-free (SynFlow) | N/R |
| Training-free (ExpressFlow) | N/R |
| Warmup (NASWOT) | 234 |
| Warmup (SNIP) | 464 |
| Warmup (SynFlow) | 289 |
| Warmup (ExpressFlow) | 227 |
| Move-Proposal (NASWOT) | 9503 |
| Move-Proposal (SNIP) | 8659 |
| Move-Proposal (SynFlow) | 9106 |
| Move-Proposal (ExpressFlow) | 6940 |
| Filtering + Refinement (ExpressFlow +Full Training) | 329 |
| ATLAS | 62 |

## F.2 COMPARISON WITH ONE-SHOT NAS METHODS

One-shot NAS approaches have shown great promise in the computer vision tasks and NAS benchmarks Liu et al. (2019b); Pham et al. (2018). They reduce search costs by training only one supernet to approximate the performance of every subnet in the search space via weight-sharing. The effective-

ness of one-shot NAS relies heavily on the correlation between the estimated performance of subnets and their true performance, where a high correlation indicates that the weights inherited from the supernet can be used for evaluation, and thus the costly training of subnets can be avoided.

Given the DNN-based search space, we train a DNN supernet comprising four layers, each with a maximum of 512 neurons. From this supernet, we evaluate the performance of subnets estimated using neurons and their weights which are inherited from the supernet. We also evaluate the actual performance of these subnets by full training. Table 12 shows the SRCC between the weight-sharing performance and the actual performance. We can find that the weight-sharing approach has a low correlation compared to our training-free metric, which leads to worse search performance.

Table 12: SRCC of Weight-Sharing and ExpressFlow on Frappe.

| Algorithm | SRCC |
|---|---|
| Weight-Sharing | 0.12 |
| ExpressFlow | 0.80 |

# G    ADDITIONAL THEORETICAL ANALYSIS

## G.1    TRAILER OVERVIEW

**Comparison of TRAILERs.** We provide the full summarization of different TRAILERs in Table 13, with their evaluation metrics, complexity, score computation definition, and the characterized properties of DNN.

Table 13: A Comparison of Different Training-Free Architecture Evaluation Metrics (TRAILERs).

| TFMEM | Evaluation Metric | Complexity | Computation | Agnostic | Property |
|---|---|---|---|---|---|
| GradNorm | Frobenius norm | 1FC+1BC | $s_a = \|\frac{\partial \mathcal{L}}{\partial \boldsymbol{\theta}}\|_F$ | Not | Expressivity |
| NASWOT | Hamming distance | 1FC | $s_a = log\|K_H\|$ | Label | Expressivity |
| NTKCond | Neural tangent kernel | 1FC+1BC | $s_a = \frac{\lambda_{max}(\Theta)}{\lambda_{min}(\Theta)}$ | Not | Trainability |
| NTKTrace | Neural tangent kernel | 1FC+1BC | $s_a = \|\Theta\|_{trace}$ | Not | Trainability |
| NTKTrace Appx | Neural tangent kernel | 1FC+1BC | $s_a = \|\Theta_{appx}\|_{trace}$ | Not | Trainability |
| Fisher | Hadamard product | 1FC+1BC | $s_a = \sum_{l=1}^{L}(\frac{\partial \mathcal{L}}{\partial ac_l}ac_l)^2$ | Not | Trainability |
| GraSP | Hessian vector product | 1FC+1BC | $s_a = \sum -(H\frac{\partial \mathcal{L}}{\partial \boldsymbol{\theta}}) \odot \boldsymbol{\theta}$ | Not | Trainability |
| SNIP | Hadamard product | 1FC+1BC | $s_a = \sum \|\frac{\partial \mathcal{L}}{\partial \boldsymbol{\theta}} \odot \boldsymbol{\theta}\|$ | Not | Trainability |
| SynFlow | Hadamard product | 1FC+1BC | $s_a = \sum \frac{\partial \mathcal{L}}{\partial \boldsymbol{\theta}} \odot \boldsymbol{\theta}$ | Data Label | Trainability |
| WeightNorm | Frobenius norm | 1FC | $s_a = \|\boldsymbol{\theta}\|_F$ | Data Label | Expressivity |
| ExpressFlow | Hadamard product | 1FC+1BC | $s_a = \sum_n \|\frac{\partial \mathcal{L}}{\partial z_n}\| \odot z_n$ | Data Label | Expressivity Trainability |

$\mathcal{L}$: loss function. $\boldsymbol{\theta}$: architecture parameters. $\Theta$: NTK matrix of the architecture.
$\odot$: Hadamard product. $s_a$: the score of a architecture $a$.
$\lambda$: the eigenvalue of NTK matrix. $H$: Hessian vector. $L$: number of architecture layers.
$\|.\|_F$: Frobenius norm. $ac_l$: activation saliency of one layer.
$FC$: forward computation. $BC$: backward computation.

**Efficiency of TRAILERs** As for efficiency, we provide both the theoretical time complexity and the average time required to score one architecture on the Frappe dataset using various TRAILERs. The experiments are run on an NVIDIA RTX 3090 and the results are presented in Table 14.

From the table, we can have the following observations. ExpressFlow is designed to be computationally efficient with only one forward and one backward computation on one mini-batch, which is the same order of magnitude as other TRAILERs. In particular, ExpressFlow takes on average $2.70 \times 10^{-3}$ seconds to score an architecture, i.e., 22,870x speedups compared to training-based architecture evaluation (61.75 seconds to fully train one architecture on the Frappe dataset).

We further evaluate their efficiency via search costs. Utilizing RE as our strategy, we iteratively seek higher-scored architectures based on TRAILERs, and monitor the time taken until we obtain one with the target AUC. The search cost for an AUC of 0.9793 on the Frappe dataset is detailed in Table 14. The results exhibit a strong correlation between the search cost and the average rank. Notably, ExpressFlow reaches the specified AUC most rapidly, outpacing the second-best metric by a factor of 6.71x. Meanwhile, many other metrics fail to achieve the target AUC. This comparison confirms the efficiency and consistency of ExpressFlow as a training-free metric in the search.

Table 14: Effiency of TRAILERs measured on Frappe dataset. FC/BC: Forward/Backward Computation, N/R: the target AUC is not reached. Search costs are evaluated on Frappe.

| | Grad Norm | NAS WOT | NTK Cond | NTK Trace | NTK TrAppx | Fisher | GraSP | SNIP | SynFlow | **ExpressFlow** |
|---|---|---|---|---|---|---|---|---|---|---|
| Computational Cost (Sec) | $2.69\times 10^{-3}$ | $1.76\times 10^{-3}$ | $5.75\times 10^{-3}$ | $5.66\times 10^{-3}$ | $2.85\times 10^{-3}$ | $9.59\times 10^{-3}$ | $1.14\times 10^{-2}$ | $3.02\times 10^{-3}$ | $2.27\times 10^{-3}$ | $2.70\times 10^{-3}$ |
| Search Cost for AUC 0.9793 (Sec) Filtering-Only | 19.9 | N/R | N/R | N/R | N/R | N/R | N/R | 18.8 | 22.1 | **2.8** |
| Search Cost for AUC 0.9798 (Sec) Filtering+Refinement | 695 | 1188 | N/R | 7568 | 4178 | 1233 | 1141 | 726 | 1184 | **62** |

**Transferability of TRAILERs**. We further examine the transferability of various TRAILERs by assessing their correlation on different data types.

We choose four widely-adopted benchmarks on image data from NAS-Bench-Suite-Zero Krishnakumar et al. (2022), i.e., NB201-CF10, NB201-CF100, NB201-IM, and NB101-CF10. For each benchmark, we evaluate the SRCC of all TRAILERs and report the results in Table 15 We find that ExpressFlow achieves the second highest average rank of SRCC in the image data, which confirms the transferability and effectiveness of ExpressFlow. In the meantime, it is a better metric to characterize *neuron saliency* of DNNs for tabular data.

Table 15: SRCC of TRAILERs Measured on Image Data

| | Grad Norm | NAS WOT | NTK Cond | NTK Trace | NTK TrAppx | Fisher | GraSP | SNIP | SynFlow | **ExpressFlow** |
|---|---|---|---|---|---|---|---|---|---|---|
| NB101-CF10 | -0.34 | 0.37 | -0.28 | -0.42 | **-0.53** | -0.37 | 0.14 | -0.27 | 0.39 | 0.38 |
| NB201-CF10 | 0.64 | 0.78 | -0.48 | 0.37 | 0.34 | 0.38 | 0.53 | 0.64 | 0.78 | **0.79** |
| NB201-CF100 | 0.64 | **0.80** | -0.39 | 0.38 | 0.38 | 0.38 | 0.54 | 0.63 | 0.76 | 0.77 |
| NB201-IM | 0.57 | **0.78** | -0.41 | 0.31 | 0.36 | 0.32 | 0.52 | 0.57 | 0.75 | 0.75 |
| Avg Rank (Image) | 4.75 | **2.0** | 7.25 | 7.25 | 7.0 | 8.25 | 7.0 | 6.0 | 3.0 | 2.5 |

### G.2 PSEUDOCODE FOR FILTERING AND REFINEMENT PHASES

In Algorithm 1 and Algorithm 2, we provide the pseudocode of the filtering and refinement phases introduced in Sections 3.2 and 3.3.

---

**Algorithm 1:** Filtering phase - Efficient Exploration

**Input** : $\mathcal{A}$: Search Space, $T_{max}$: Time budget. $Q$: Message Queue. $X_B$: Mini-batch of data. $\mathcal{N}$: Number of architecture encoding.

**Output**: $K$ architecture.

1   **Aging Evolution** *($\mathcal{A}$, $T_{max}$)*

     /* Randomly generate a pool of $\mathcal{N}$ encoding */

2      $Pool \leftarrow$ RandomArchitecture($\mathcal{N}$)

     /* Thread-1 */

3      **while** *Receive worker's request* **do**

4          encoding $\leftarrow$ GeneticSelection($Pool$)

5          encoding' $\leftarrow$ MUTATE(encoding)

6          $Q \leftarrow Q \bigcup$ (encoding')

7      **while** *True* **do**

         /* Fetch (architecture, score) from queue*/

8          (encoding, $s_a$) $\leftarrow$ $Q$.FetchScore()

9          UpdateLocalPool(encoding, $s_a$)

10   **Evaluation Worker** *($X_B$)*

11      **while** *True* **do**

         /* Score the architecture */

12          encoding' $\leftarrow$ $Q$.FetchArchitecture()

         /* Construct the architecture */

13          $a \leftarrow$ ArchitectureConstruct(encoding')

14          $s_a \leftarrow \rho(a, \boldsymbol{\theta}, X_B)$

15          $Q \leftarrow Q \bigcup$ (encoding', $s_a$)

---

**Algorithm 2:** Refinement phase - Effective Exploitation

**Input** : $K$: Top architectures. $U$: Min computation unit. $\eta$: $\frac{K}{\eta}$ architectures to keep per-round.

**Output**: The final selected architecture.

1   **Training Worker** *(K)*

2      $U_{cur} \leftarrow U$

3      **while** $K.length() > 1$ **do**

         /* Evaluate each architecture using $U_0$ epoch */

         $\{p_{a_i}; a_i \in K\} \leftarrow \{Eval(a_i, U_{cur}); a_i \in K\}$

         /* Only keep top $\frac{K}{\eta}$ architectures */

         $K \leftarrow Top_{\frac{1}{\eta}}(\{p_{a_i}; a_i \in K\})$

         /* Increase epoch-per-architecture in next round */

         $U_{cur} \leftarrow U_{cur} \cdot \eta$

---

### G.3 THEORETICAL ANALYSIS OF EXPRESSFLOW

As defined in Section 3.2, the neuron saliency for the $n$-th neuron $\nu_n$ is defined as follows:

$$\nu_n = |\frac{\partial \mathcal{L}}{\partial z_n}| \bigodot z_n \tag{3}$$

where $z_n = \sigma(h_n)$ and $h_n = \mathbf{w}\mathbf{x} + b = \sum_u \mathbf{w_{nu}^{in}} z_u + b$. $\mathbf{w_{nu}^{in}}$ represents input parameter between the $n$-th neuron and $u$-th neuron in the preceding layer.

Further, we denote $\mathbf{w_{vn}^{out}}$ as the parameter between the $n$-th neuron and the $v$-th neuron in the following layer. We denote the neuron index in the following layer as $i$. Then we can have

$$\frac{\partial \mathcal{L}}{\partial z_n} = \sum_i \frac{\partial \mathcal{L}}{\partial h_v} \frac{\partial h_v}{\partial z_n} \tag{4}$$

given $h_v = \sum_n \mathbf{w_{vn}^{out}} z_n + b$, we can have:

$$\frac{\partial h_v}{\partial z_n} = \mathbf{w_{vn}^{out}} + \sum_{k \neq n} \frac{\partial \mathbf{w_{vk}^{out}} z_k}{\partial z_n} = \mathbf{w_{vn}^{out}} \tag{5}$$

and

$$\frac{\partial h_v}{\partial \mathbf{w_{vn}^{out}}} = z_n + \sum_{k \neq n} \frac{\partial \mathbf{w_{vk}^{out}} z_k}{\partial \mathbf{w_{vn}^{out}}} = z_n \tag{6}$$

then we get

$$\frac{\partial \mathcal{L}}{\partial z_n} = \sum_i \frac{\partial \mathcal{L}}{\partial h_v} \mathbf{w_{vn}^{out}} \tag{7}$$

The neuron saliency $\nu_n$ can be represented as

$$\nu_n = |\sum_i \frac{\partial \mathcal{L}}{\partial h_v} \mathbf{w_{vn}^{out}}| \bigodot z_n \tag{8}$$

As for the homogeneous activation functions $\sigma$ including ReLU and Leaky ReLU, $z_n = \sigma(h) \geq 0$. And for any specific neuron $n$, the Hadamard product is the same as the simple multiplication.

$$
\begin{aligned}
\nu_n &= |\sum_i \frac{\partial \mathcal{L}}{\partial h_v} \mathbf{w_{vn}^{out}} z_n| \\
&= |\sum_i (\frac{\partial \mathcal{L}}{\partial h_v} z_n) \mathbf{w_{vn}^{out}}|
\end{aligned}
\tag{9}
$$

With equation 6, we have

$$
\begin{aligned}
\nu_n &= |\sum_i (\frac{\partial \mathcal{L}}{\partial h_v} \frac{\partial h_v}{\partial \mathbf{w_{vn}^{out}}}) \mathbf{w_{vn}^{out}}| \\
&= |\sum_i \frac{\partial \mathcal{L}}{\partial \mathbf{w_{vn}^{out}}} \mathbf{w_{vn}^{out}}|
\end{aligned}
\tag{10}
$$

Evidently, $\frac{\partial \mathcal{L}}{\partial \mathbf{w_{vn}^{out}}} \mathbf{w_{vn}^{out}}$ aligns precisely with the concept of synaptic saliency, as in the referenced literature Tanaka et al. (2020). This measure was originally proposed to assess the individual significance of each parameter $\mathbf{w_{vn}^{out}}$.

Based on the equation 10, the neuron saliency of the $n$-th neuron actually computes the absolute value of the sum of SynFlow scores of all outgoing parameters of each neuron. Compared with the SynFlow, ExpressFlow calculates saliency on a neuron-wise basis. Further, the use of absolute value computation prevents mutual cancellation of neuron saliency, which ensures the importance of all neurons is captured. Our ExpressFlow thus provides a more accurate and consistent evaluation of DNN architectures for tabular data, as analyzed in Section 3.2.

Moreover, homogeneous activation functions $\sigma$ indicates $z_n = \sigma(h) = \sigma(h)^{'} h$. For any specific neuron $n$, we have

$$
\begin{aligned}
\nu_n &= |\frac{\partial \mathcal{L}}{\partial z_n}| \bigodot z_n = |\frac{\partial \mathcal{L}}{\partial h_n}| \bigodot h_n = |\frac{\partial \mathcal{L}}{\partial h_n} h_n| \\
&= |\frac{\partial \mathcal{L}}{\partial h_n} \sum_u (\mathbf{w_{nu}^{in}} z_u + b)| \\
&= |\sum_u (\frac{\partial \mathcal{L}}{\partial h_n} \mathbf{w_{nu}^{in}} z_u + \frac{\partial \mathcal{L}}{\partial h_n} b)| \\
&= |\sum_u (\frac{\partial \mathcal{L}}{\partial h_n} \frac{\partial h_n}{\partial \mathbf{w_{nu}^{in}}} \mathbf{w_{nu}^{in}} + \frac{\partial \mathcal{L}}{\partial h_n} b)| \\
&= |\sum_u (\frac{\partial \mathcal{L}}{\partial \mathbf{w_{nu}^{in}}} \mathbf{w_{nu}^{in}} + \frac{\partial \mathcal{L}}{\partial h_n} b)| \\
&= |\sum_u (\frac{\partial \mathcal{L}}{\partial \mathbf{w_{nu}^{in}}} \mathbf{w_{nu}^{in}}) + \sum_u \frac{\partial \mathcal{L}}{\partial h_n} b|
\end{aligned}
\tag{11}
$$

Interestingly, the term $\frac{\partial \mathcal{L}}{\partial \mathbf{w_{nu}^{in}}} \mathbf{w_{nu}^{in}}$ also signifies the synaptic saliency correlated with the input parameter $\mathbf{w_{nu}^{in}}$ of the $n$-th neuron. Additionally, the computation of the neuron saliency $\nu_n$ encompasses the aggregate of its respective input synaptic saliencies, which effectively captures the complex and non-intuitive relationships among input features in tabular data.

Finally, we aggregate all neuron saliency of each layer as the final score of the given architecture $a$. With equation 11, we have

$$
\begin{aligned}
s_a &= \sum_{i=1}^{B} \sum_{n=1}^{N} \nu_{in} = \sum_{i=1}^{B} \sum_{n=1}^{N} |\frac{\partial \mathcal{L}}{\partial z_{in}}| \bigodot z_{in} \\
&= \sum_{i=1}^{B} |\sum_{n=1}^{N} \sum_u (\frac{\partial \mathcal{L}}{\partial \mathbf{w_{nu}^{in}}} \mathbf{w_{nu}^{in}}) + \sum_n \sum_u \frac{\partial \mathcal{L}}{\partial h_n} b|
\end{aligned}
\tag{12}
$$

which measures both the trainability and expressivity of the architecture $a$.

### G.4 DATA-AGNOSTIC PROPERTIES OF EXPRESSFLOW

In this paper, we exclusively employ the ReLU activation function, with $z_n = \sigma(h) = 0$ holding for any $h \leq 0$. This particular choice inherently implies that certain neurons may yield a zero neuron saliency, given that the saliency is defined by $\nu_n = |\frac{\partial \mathcal{L}}{\partial z_n}| \bigodot z_n$. Consequently, this could potentially result in the total score not fully encapsulating the information provided by the neurons, thus leading to a diminution in the correlation of the ExpressFlow. This is substantiated by empirical studies presented in Appendix D.1.

We, therefore, restrict to the positive value of $h$ by requiring that each non-zero parameter is positive, i.e., we set $\mathbf{w} = |\mathbf{w}|$ for all $\mathbf{w} \in a$. Specifically, for $h = \mathbf{w}z + b$, given $\mathbf{w}$ is positive and $z$ is non-negative, $h$ could keep positive for all non-zero $z$.

Empirical findings shown in Appendix D.1 suggest that the correlation remains largely unaffected by different batch sizes. Based on this observation, we define a novel loss function to aid in the computation of ExpressFlow:

$$\mathcal{L}(X_B) = a(\mathbf{1}_d) = z_{L+1}. \tag{13}$$

where $z_{L+1}$ denotes the output of the final layer of the architecture, $\mathbf{1}_d$ represents an all-one vector of dimension $d$ and batch size $B$, and $a(\cdot)$ is the architecture mapping this vector to the output $z_{L+1}$.

In conclusion, the computation of ExpressFlow is both efficient and data-agnostic, which significantly simplifies the process of architecture evaluation during the filtering phase, which subsequently facilitates anytime NAS on various tabular datasets.

# H  NOTATION AND TERMINOLOGY

Table 16: Summary of Notation and Terminology

| | |
|---|---|
| $T_{max}$ | Time budget |
| $\mathcal{A} = \{a\}$ | Search space and architecture |
| $L$ | Number of layers in the DNN |
| $\mathcal{H}$ | A candidate set of layer sizes |
| $|\mathcal{H}|^L$ | Number of candidate architectures |
| $f_s$ | Search strategy |
| $\mathcal{S}_i$ | State of the search strategy at the $i$-th iteration. |
| $\boldsymbol{\theta}$ | Parameter of an architecture |
| $B, X_B$ | Batch size. A batch of data samples |
| $N$ | Number of neurons of the whole architecture |
| $N_l$ | Number of neurons of the $l$-th layer of the architecture |
| $\mathcal{L}$ | Loss function |
| $s_a$ | TRAILER score of architecture |
| $\rho(\cdot)$ | TRAILER assessment function |
| $\odot$ | Hadamard product |
| $\Theta$ | NTK metrics |
| $\Phi$ | Synaptic saliency |
| $\nu_n$ | Neuron saliency of the $n$-th neuron computed on a batch of samples |
| $\nu_{in}$ | Neuron saliency of the $n$-th neuron computed on the $i$-th sample |
| $z_n$ | Activated output of the $n$-th neuron computed on a batch of samples |
| $z_{in}$ | Activated output of the $n$-th neuron computed on the $i$-th sample |
| $\mathbf{w}$ | Incoming weights of the neuron |
| $\sigma$ | Activation function |
| $\mathbf{x}$ | Neuron inputs |
| $\mathcal{K}_l$ | Number of neurons in the $l$-th layer |
| $z^d(t)$ | Trajectory of the output of the layer $d$, and is parametrized by scalar $t$ |
| $\ell(\cdot)$ | Trajectory length |
| $\ell(z^d(t))$ | Trajectory length of the output of the layer $d$ |
| $M$ | Number of architectures explored in the filtering phase |
| $K$ | Number of architectures exploited in the refinement phase |
| $U$ | Computational unit in the refinement phase |
| $t_1$ | Time to score an architecture |
| $t_2$ | Time to train an architecture for a single epoch |
| $T_1$ | Time allocated to the filtering phase |
| $T_2$ | Time allocated to the refinement phase |
| $H$ | Hessian vector |
| $\mathcal{N}$ | Number of architecture encodings |
| $\eta$ | $\frac{K}{\eta}$ Architectures to keep per-round |

# I  CODE AND DATA

For reproducibility and comparisons in NAS research, we have made the source code available at the following URL: https://anonymous.4open.science/r/ATLAS/ This repository contains the experimental code, the experimental data, and a detailed guide outlining the steps to reproduce the results, located within the README.md file. Furthermore, the link to NAS-Bench-Tabular is also provided in the same README.md file.

