# OpenReview forum: "Anytime Neural Architecture Search on Tabular Data"
_ICLR.cc/2024/Conference — Submitted to ICLR 2024_

### Official Review · Reviewer_wdfZ · 2023-10-31

**Soundness:** 3 good
**Presentation:** 3 good
**Contribution:** 2 fair
**Rating:** 6
**Confidence:** 4

**Summary:**

The contribution of paper consists of three works. First, they construct NAS-Bench-Tabular, a DNN-based search space for tabular data. Second, they propose a novel performance proxy of neural network, ExpressFlow. ExpressFlow is based on neuron saliency, but also reflects the difference of importance with depth. Finally, they propose ATLAS, a NAS approach to tabular data.

**Strengths:**

To the best of my knowledge, this paper is the first NAS benchmark for tabular data. This work will reduce unnecessary redundant experimentation with NAS for tabular data. The proposed ExpressFlow and ATLAS show better performance than the existing Tabular NAS method despite having a relatively simple structure.

**Weaknesses:**

The search space of NAS-Bench-Tabular(5-layer DNN with different width) is too simple. This search space do not contain most of modern neural network architecture for tabular data[1].

Most of the candidate architectures get similar 97~98% accuracy on Frappe dataset. I wonder how stable the recorded accuracies are.

All three datasets are about binary classification tasks. Regression, Multi-class classification tasks are needed.

DNNs are not the dominant methodology for tabular data, and machine learning techniques such as XGBoost and CatBoost have shown competitive performance [1]. With a limited search space for DNNs, it's hard to see how ATLAS has an advantage over models like AutoSklearn[2] and AutoGluon[3].


[1] Gorishniy, Y., Rubachev, I., Khrulkov, V., & Babenko, A. (2021). Revisiting deep learning models for tabular data. Advances in Neural Information Processing Systems, 34, 18932-18943.

[2] Feurer, M., Eggensperger, K., Falkner, S., Lindauer, M., & Hutter, F. (2022). Auto-sklearn 2.0: Hands-free automl via meta-learning. The Journal of Machine Learning Research, 23(1), 11936-11996.

[3] Erickson, N., Mueller, J., Shirkov, A., Zhang, H., Larroy, P., Li, M., & Smola, A. (2020). Autogluon-tabular: Robust and accurate automl for structured data. arXiv preprint arXiv:2003.06505.

**Questions:**

In general, performance proxies have the advantage of being data type-agnostic, which is also a benefit of deep learning. Why do we need a TRAILER that specializes in tabular data? And what makes ExpressFlow specializes on tabular data?

In Equation 2, aren’t Successive Halving Algorithm allocates different budget U and candidate K for each round?

---

> ### Author Response · Authors · 2023-11-22
>
> We thank you for your constructive comments and positive feedback. We would like to address your concerns below.
>
> >The search space of NAS-Bench-Tabular ... tabular data[1].
>
> **Response:** Existing studies show that deep neural networks (DNNs) can already achieve state-of-the-art performance on tabular data [4, 5], and the technical challenge is to configure DNNs with the right number of layers and hidden layer sizes for each layer [1]. Therefore, we design a DNN-based search space that comprises up to 160,000 candidate DNNs with an extensive collection of different layer configurations.
>
> As shown in the experiments and visualizations in Section 4.1.1 and Appendix C.3, this search space contains a diverse set of architectures with a wide range of parameter sizes, configurations, and performances, which captures the main properties of DNNs for benchmarking NAS approaches on tabular data.
> > Most of the candidates... stable the recorded accuracies are.
>
> **Response:** We have summarized the statistics of our 160,000 trained architectures on the Frappe dataset in the table below.
>
> | Mean (AUC) | Standard Deviation (AUC) | Maximum (AUC) | Minimum (AUC) |
> | - | - | - | - |
> | 97.68%     | 0.0021                   | 98.14%        | 85.72%        |
>
> The very low standard deviation (0.0021) indicates that most architectures closely align with the mean AUC. While the significant gap between the maximum AUC (98.14\%) and minimum AUC (85.72\%) suggests that the number of hidden neurons in each layer is crucial in determining the architecture performance.
> > All three datasets are about ... multi-class classification tasks are needed.
>
> **Response:** As suggested, we evaluated ATLAS on additional datasets: three for multi-class classification and two for regression.
> The results of multi-class classification are presented in our global response to the _Evaluation of ATLAS on more Datasets_.
> The results of the regression are shown in the table below where we have measured the Root Mean Square Error (RMSE).
>
> | **Dataset**         | **XGBoost** | **CatBoost** | **ATLAS** | **Searched Architecture** |
> | - | - | - | - | -|
> | California Housing  | 0.139       | **0.131**    | 0.141     | 384-256-256-512           |
> | Pol                    | 4.773       | 4.698        | **2.264** | 384-256-128-256           |
>
> ATLAS outperforms both XGBoost and CatBoost on Pol datasets with lower RMSE.
> While for the California Housing dataset, ATLAS achieves comparable results with others.
> >DNNs are not the dominant methodology ...  AutoGluon[3].
>
> **Response:** As suggested, we have compared ATLAS with XGBoost, CatBoost, AutoSklearn, and AutoGluon on eight datasets.
> The results are presented in our global response to the _Evaluation of ATLAS on more Datasets_.
>
> > In general, performance proxies ... And what makes ExpressFlow specialize in tabular data?
>
> **Response:** Existing data type-agnostic TRAILERs such as SynFlow do not perform well on tabular datasets as shown in Table 1 of the original paper. Therefore, We are motivated to propose a new TRAILER specifically designed for tabular data.
>
> To this end, we first characterize the trainability and expressivity of existing training-free metrics initially proposed for image data. Then, we propose a new TRAILER tailored for tabular data based on neuron saliency.
> This TRAILER characterizes both properties for DNNs on tabular data (see discussion in Section 3.2 and theoretical analysis in Appendix D.3), by considering the activation value of neurons that are basic units to extract features in DNNs and the derivatives of neurons that indicate the importance of features extracted by neurons.
>
> > In Equation 2, ... candidate K for each round?
>
> **Response:** Our ATLAS includes both filtering and refinement phases. For the refinement phase, it uses the Successive Halving Algorithm (SUCCHALF) to identify the best-performing architecture from the $K$ architectures, with each training for at least $U$ epochs.
>
> Indeed, SUCCHALF allocates a different budget of $U * \eta^i$ (with $U$ doubling if $\eta$ = 2) and a number of candidate architectures $\lfloor K/\eta^i \rfloor$ (halving if $\eta$ = 2) for training-based evaluation in the $i$-th round (with i starting from 0).
> For example, in the first round of evaluation, SUCCHALF trains $K$ architectures, with each training for $U$ epochs.
> In the second round, the higher-performing $\lfloor K/\eta\rfloor$ architectures are retained, with each training for $U * \eta$ epochs. This process iterates for $\lfloor\log_{\eta}K\rfloor$ training rounds until one single architecture remains.
>
> We hope our responses above have addressed your concerns and can improve your evaluation of our work.
>
> [1] Revisiting Deep Learning Models for Tabular Data [Gorishniy et al., NeurIPS 2021].
>
> [4] Well-tuned simple nets excel on tabular datasets [Kadra et al., NeurIPS 2021].
>
> [5] TabNAS: Rejection Sampling for Neural Architecture Search on Tabular Datasets [Yang et al., NeurIPS 2022].

---

### Official Review · Reviewer_uv1h · 2023-11-02

**Soundness:** 3 good
**Presentation:** 4 excellent
**Contribution:** 3 good
**Rating:** 6
**Confidence:** 5

**Summary:**

This paper proposes a new NAS algorithms for tabular data, which is based on both zero-cost proxies for distinguishing between good and bad architectures, as well as a multi-fidelity method (Successive Halving) for allocating budget to these promising architectures selected in the first phase. Another contribution of this paper is the creation of NAS-Bench-Tabular, which comprises a search space with a total of 160k unique trained and evaluated architectures. A new zero-cost proxy, tailored for tabular data, is theoretically derived and is compared to existing zero-cost proxies previously designed for vision tasks, exhibiting better performance in the selected benchmarks.

**Strengths:**

- The motivation to work on anytime NAS algorithms for tabular data is valid considering the significance of the problem and domain.

- The *ExpressFlow* zero-cost proxy (ZCP) is specifically tailored to the tabular modality and is justified by a thorough theoretical derivation and later backed by empirical results.

- The authors propose and construct a new NAS benchmark for tabular data, named NAS-Bench-Tabular. The search space is exhaustively evaluated over 3 datasets. This is certainly very useful for the community, as the previous NAS benchmarks have been, and I am certain it will accelerate the research speed on NAS for tabular data.

- The proposed ZCP is fairly compared to other state-of-the-art ZCPs that were designed mainly for the image domain and the respective architectures, and the results show that *ExpressFlow* outperforms them by a considerable margin.

- The idea to incorporate *ExpressFlow* into regularized evolution to evolve a population of architectures that optimize the ZCP and then run SuccessiveHalving to allocate budget to more promising configurations inside the evolved population is simple and effective.

- The paper is easy to follow and I really enjoyed reading it. The experimental setup and results are demonstrated clearly.

- The authors provide the code necessary to reproduce the results in the paper.

**Weaknesses:**

Despite the strengths I mentioned above, there are some really important points that have to be addressed, especially regarding the empirical evaluation.

- **Benchmark needs to be more diverse for practical purposes**: I think that the creation of NAS-Bench-Tabular is very useful, however when it comes to tabular data, a lot of datasets contain just a handful of training examples (less than 1000), and more features than the datasets the authors chose. Refer to [1] (Table 9 in the appendix) for an example.

- **Evaluation of ATLAS on more diverse datasets**: This paper contains multiple contributions, starting from the proposal of a new NAS benchmark for tabular data to the NAS algorithm for tabular data. While these are very useful and provide interesting insights, it can also introduce many biases. Typically, one would design an algorithm that works on benchmarks (real ones) that the community already uses in their research. Firstly designing a NAS benchmark for tabular data and then proposing a method and evaluating it only on that benchmark does not guarantee the same behavior on the real benchmarks. Therefore, I would really be interested to see how ATLAS works on commonly used OpenML tabular benchmarks (e.g. the ones from [1], Table 9), that are more diverse and lie more into the small-data regime. If the authors provide competitive results on those datasets, I am willing to increase my score.

- **Comparison with conventional tree-based models**: To assess the usefulness of ATLAS in practice, I think the authors should compare the best found network with conventional tree-based models, as it is done in [1], [2] and [3], for instance. An easy and fair experimental setup would be: (1) Run ATLAS for a given time budget to find architecture **x**; (2) Train and evaluate **X** on the full fidelity (if not already evaluated in (1)); (3) Run tree-based models for *t* time (total runtime of (1) + runtime of (2)).

- **Comparison to SOTA for DL for tabular data**: A thorough comparison to neural networks that achieve SOTA for tabular data is necessary to highlight the usefulness of NAS in this domain. Similar to the point above I would recommend the authors to compare to TabPFN [3]. If the time permits, I would recommend the authors to run ATLAS on the same settings the TabPFN authors used to obtain Figure 5 in their paper.

**Minor**

- It would be great if the authors follow the already used nomenclature when referring to zero-cost proxies, and not rename them to TRAILERs.

- In Section 2, third paragraph, it is written "The performance obtained by the training-based architecture evaluation approaches is accurate", however this is not always the case, especially when considering proxy models (less layers, channels, etc.) or performance evaluation based on the one-shot model weights (e.g. as in DARTS).

- Most of blackbox algorithms are also anytime algorithms, and they are widely used for NAS as well. Therefore, the claim that ATLAS is the first NAS anytime algorithm (in abstract) for tabular data is wrong considering that the architectural parameters have been optimized using these methods (e.g. Bayesian Optimization) and could easily be used to optimize architectural hyperparameters for tabular data networks as well. Ideally, one wants an optimization method that is modality agnostic. ATLAS is the first anytime algorithm tailored for tabular data because of *ExpressFlow* is designed for tabular data. And yes, *ExpressFlow* is the first zero-cost proxy designed for tabular data, so I would emphasize that instead.

**References**

[1] https://arxiv.org/pdf/2106.11189.pdf

[2] https://arxiv.org/pdf/2106.11959.pdf

[3] https://arxiv.org/pdf/2207.01848.pdf

**Questions:**

- Why did the authors pick those 3 datasets for their benchmark? Is it because of the large number of training examples?

- Is ATLAS performant on datasets with less than 1000 data points?

- How many data points were used to compute the rank correlation in Table 1?

---

> ### Author Response · Authors · 2023-11-22
>
> We appreciate your valuable feedback and constructive comments on helping improve our paper. We would like to address your concerns below.
> >Benchmark needs to be more diverse ... for an example.
>
> >Evaluation of ATLAS on more diverse datasets ... increase my score.
>
> >Is ATLAS performant on datasets with less than 1000 data points?
>
> **Response:** As suggested, we have evaluated our ATLAS on more datasets from the open-source OpenML AutoML Benchmark, and the results are presented in our global response to the _Evaluation of ATLAS on more Datasets_.
> > Comparison with conventional tree-based models ...  runtime of (2)).
>
> **Response:** As suggested, we compared ATLAS with XGBoost, assessing their AUC on the Frappe dataset.
>
> Our ATLAS, supporting anytime NAS for tabular data, can operate within a pre-defined time budget.
> We first set a time budget to run ATLAS to select a well-performing architecture. Then, an additional time budget was allocated to fully train and evaluate this selected architecture.
> We managed both time budgets to ensure the total time budget was around one hour.
>
> For XGBoost, we also fixed the overall time budget for one hour and tuned the hyperparameter of XGBoost based on training. We employed the same hyperparameter search space as defined in [1] (Table 6).
>
> The results shown in the Table below demonstrate that ATLAS can identify architectures that outperform XGBoost on the Frappe datasets. This confirms the efficacy of ATLAS in finding well-configured DNNs, i.e., determining the appropriate number of hidden neurons for each layer.
>
> | **Dataset** | **#Samples/#Columns** | **XGBoost** | **ATLAS**  |
> | - | - | - | - |
> | Frappe      | 288,609 / 10          | 97.46%      | **98.03%** |
> > Comparison to SOTA for DL for tabular data ... obtain Figure 5 in their paper.
>
> **Response:** Our ATLAS differs theoretically from TabPFN.
> TabPFN is a Prior-Data Fitted Network (PFN) capable of approximating probabilistic inference in a single forward pass, it has two limitations: 1) it requires extensive offline training (e.g., 20 hours on 8 GPUs as reported in [3]), and 2) it only scales effectively to small datasets and performs worse when categorical features are presented.
>
> In contrast, our ATLAS employs both training-free and training-based architecture evaluation techniques.
> When compared to TabPFN, ATLAS demonstrates robust performance across diverse datasets ranging from small to large and consistently performs well when facing either numerical or categorical features, as evidenced in Figure 6 of the original paper.
>
> Table 5 in the TabPFN paper [3] presents the mean AUC of the TabPFN across all 18 datasets from the OpenML-CC18 Benchmark. Due to time constraints, we have evaluated our ATLAS on only eight datasets. For each dataset, we have also measured the total time (Total Cost) it takes to both search for the higher-performing architecture and then train this architecture to evaluate its performance, as shown in our global response to the _Evaluation of ATLAS on more Datasets_.
> The results illustrate ATLAS is highly efficient and is competitive with methods detailed in Table 5 of the TabPFN paper.
> > It would be great if the authors ... and not rename them to TRAILERs.
>
> **Response:** As suggested, we will refer the TRAILERs to zero-proxies.
> > In Section 2, third paragraph ... performance evaluation based on the one-shot model weights (e.g. as in DARTS).
>
> **Response:** As suggested, we will rephrase the sentence to "The performance obtained by the training-based architecture evaluation approaches is typically more accurate".
> > Most of blackbox algorithms are also anytime algorithms,  ... so I would emphasize that instead.
>
> **Response:** Thank you for your valuable suggestions. We will first rephrase our claims to state 'ATLAS is the first anytime algorithm tailored for tabular data.' Second, we will emphasize that 'ExpressFlow is the first zero-cost proxy designed for tabular data' in our paper.
> > Why did the authors pick those 3 datasets for their benchmark? Is it because of the large number of training examples?
>
> We explain the reason for choosing the three datasets in our global response to _Reasons for Selecting the Three Datasets for NAS-Bench-Tabular_.
> > How many data points were used to compute the rank correlation in Table 1?
>
> **Response:** In Table 1 of our paper, we present the correlation between each training-free evaluation metric score and the architecture's actual performance (AUC).  This analysis uses 160,000 architectures for the Frappe and Diabetes datasets and 10,000 for the Criteo dataset.
>
> We hope our responses above have addressed your concerns and can improve your evaluation of our work.
>
> [1] Well-tuned simple nets excel on tabular datasets [Kadra et al., NeurIPS 2021].
>
> [2] Revisiting Deep Learning Models for Tabular Data [Gorishniy et al., NeurIPS 2021].
>
> [3] TabPFN: A Transformer That Solves Small Tabular Classification Problems in a Second [Noah et al., ICLR 2023].

---

> > ### Comment · Reviewer_uv1h · 2023-11-22
> > **Thank you for your response**
> >
> > I thank the authors for putting the effort to address some of my main concerns. I will increase my score and suggest the authors to update the main paper with the new results in their global response + additional results on the remaining benchmarks of OpenML-CC18.
> >
> > One other thing that I would suggest is to change the name of the benchmark from "NAS-Bench-Tabular". This can create confusion w.r.t. tabular benchmarks (which are not necessarily on tabular data). For instance, some alternative can be "NAS-Bench-TD" (TD stands for Tabular Data).

---

> ### Author Response · Authors · 2023-11-22
> **Thank you for raising your score!**
>
> We are very glad to know that our response has addressed most of your concerns with clarity.
> We would also like to thank you for raising your score for our paper.
> As per your suggestion, we will update the main paper with the results from our global response, along with additional evaluations on the OpenML-CC18 benchmarks. Additionally, to avoid confusion and for clearer representation, we will rename "NAS-Bench-Tabular" to "NAS-Bench-TD".
> Thank you again for your constructive comments.

---

### Official Review · Reviewer_b3YS · 2023-11-05

**Soundness:** 2 fair
**Presentation:** 3 good
**Contribution:** 2 fair
**Rating:** 5
**Confidence:** 3

**Summary:**

The paper introduces an approach called ATLAS, which focuses on Anytime Neural Architecture Search (NAS) specifically designed for analyzing tabular data, an area that has not been extensively explored in NAS research. With the aim of addressing the need for efficient NAS methods that can accommodate different time constraints, ATLAS presents a two-phase optimization strategy that cleverly combines training-free and training-based evaluations.  Experimental results demonstrate ATLAS's ability to quickly deliver competent architectures while adapting to expanded time budgets. Compared to conventional NAS methods, ATLAS achieves a remarkable reduction in search times, up to 82.75 times faster. These findings highlight the effectiveness and efficiency of ATLAS in the context of tabular data analysis, showcasing its potential for accelerating the NAS process.

**Strengths:**

1) The development of ATLAS as the first Anytime NAS approach specifically for tabular data is a significant advantage and the main contribution of the paper. It fills a gap in the field of neural architecture search by providing a solution tailored to tabular datasets. ATLAS's innovative approach allows for the adaptation to any given time constraint, ensuring that the best possible architecture is available within the set time frame and can improve if more time is allocated. This responsiveness to computational budget constraints is a substantial step forward for practical applications of NAS in real-world scenarios where time and resources are often limited.

2) The paper is well-written and positioned within the existing body of literature. The paper is commended for its clear writing, which concisely explains complex technical processes. It lays out the limitations of current approaches and systematically introduces the novel contributions of ATLAS, establishing its significance in the context of NAS research. The authors have ensured that the paper is not only informative but also accessible, making it a valuable addition to the academic discourse on neural architecture search.

**Weaknesses:**

1) A limitation of the paper is the selection of benchmark datasets that may not comprehensively represent the diversity of real-world tabular data. The features of the chosen datasets—Frappe, Diabetes, and Criteo—are relatively low in dimensionality (10, 43, and 39 features, respectively). This narrow scope could potentially limit the generalizability of the study's findings. To convincingly argue the efficacy of ATLAS across various scenarios, it would be beneficial to test it on a wider range of datasets with varying feature dimensions, complexity, and domain-specific challenges. The current dataset selection might not fully challenge the capability of ATLAS to handle higher-dimensional and more complex tabular datasets that are commonly found in practice.

2) The missing significant baselines, particularly established methods such as XGBoost and various Transformer-based models like TabTransformer and FTTransformer. Including these baselines is crucial for a comprehensive comparative analysis, especially to substantiate the necessity and superiority of NAS in the domain of tabular data. By neglecting to compare ATLAS against these well-known and widely-used methods, the paper misses an opportunity to demonstrate the practical advantage of NAS for tabular data over more traditional, yet powerful, approaches. This comparative analysis is essential to persuade the research community and industry practitioners of the added value that ATLAS and, more broadly, NAS methods may provide in tabular data applications.

**Questions:**

Resolve the concerns in Weakness section.

---

> ### Author Response · Authors · 2023-11-22
>
> We highly appreciate your feedback. However, we would like to point out some misunderstandings about our work.
>
> >The development of ATLAS ... of NAS in real-world scenarios where time and resources are often limited.
>
> **Response:** Thank you for appreciating the contribution of our NAS tabular data benchmark (NAS-Bench-Tabular). We will make it publicly available to facilitate further research and development in the area of NAS for tabular data.
>
> > A limitation of the paper ... and more complex tabular datasets that are commonly found in practice.
>
> **Response:** The datasets utilized in our work are real-world tabular data. Specifically, we select three datasets from different application domains: app recommendation, healthcare analytics, and CTR (e-commerce) prediction. The statistics of the datasets are summarized in the Table below:
>
> | Dataset  | # Class | # Samples  | # Columns | # Features | Task                 |
> | -------- | ------- | ---------- | --------- | ---------- | -------------------- |
> | Frappe   | 2       | 288,609    | 10        | 5,382      | App Recommendation   |
> | Diabetes | 2       | 101,766    | 43        | 369        | Healthcare Analytics |
> | Criteo   | 2       | 45,840,617 | 39        | 2,086,936  | CTR Prediction       |
>
> Notably, for the three datasets, there are 10, 43, and 39 columns, corresponding to 5,382, 369, and  2,086,936 unique numerical and categorical features respectively, which are high-dimensional.
>
> To further validate the generalizability and effectiveness of our proposed ATLAS, we have conducted experiments on an additional eight datasets from the open-source OpenML AutoML Benchmark.
>
> The evaluation results, as presented in our global response to the _Evaluation of ATLAS on More Datasets_, confirm the effectiveness and generalizability of our proposed ATLAS.
>
> > The missing significant baselines ... NAS methods may provide in tabular data applications.
>
> **Response:** We would like to clarify that the scope of this paper is to propose a novel anytime NAS approach tailored for tabular data, rather than to design a new architecture that outperforms established architectures such as TabTransfer or FT-Transfermor.
> As a result, we mainly pick baselines from the state-of-the-art NAS approaches designed for tabular data.
> Specifically, we utilize TabNAS [1] and training-based NAS approaches, i.e., RE-NAS, for comparison as shown in Table 6 and discussed in Section 4.3 of the original paper.
>
> To further demonstrate the practical advantages of our ATLAS, we have measured its balanced accuracy on two datasets from the OpenML AutoML Benchmark and compared ATLAS against XGBoost, TabTransformer, and FTTransformer. The results shown in the table below indicate that the selected architecture of ATLAS is comparable to both transformer-based architectures and XGBoost.
>
> | **Dataset** | **#Samples/#Columns** | **XGB** | **TabTransformer** | **FTTransformer** | **ATLAS** | **Searched Architecture** |
> | ----------- | --------------------- | ------- | ------------------ | ----------------- | --------- | ------------------------- |
> | Adult       | 48842 / 15            | 79.82   | 78.09              | 78.42             | 78.40     | 384-256-256-512           |
> | Bank        | 45211 / 17            | 72.66   | 78.57              | 78.37             | 78.32     | 384-256-256-512           |
>
> Both TabTransformer and FTTransformer are designed to capture complex relationships in tabular data through self-attention mechanisms, and thus they perform consistently well on two datasets.
> Notably, our ATLAS effectively determines the number of hidden neurons for each layer of an MLP, thereby making it competitive with other methods. The experimental results align with conclusions drawn from FTTransformer [2]: specific tuning can make simple models like MLPs competitive.
>
> We hope our responses above have addressed your concerns and can improve your evaluation of our work.
>
> [1] TabNAS: Rejection Sampling for Neural Architecture Search on Tabular Datasets [Yang et al., NeurIPS 2022].
>
> [2] Revisiting Deep Learning Models for Tabular Data [Gorishniy et al., NeurIPS 2021].

---

> ### Author Response · Authors · 2023-11-23
>
> We hope our clarifications and new experiments have addressed your concerns. We would highly appreciate your consideration for re-evaluating your initial rating. We look forward to any further feedback you may have.

---

### Official Review · Reviewer_uv7V · 2023-11-07

**Soundness:** 3 good
**Presentation:** 3 good
**Contribution:** 2 fair
**Rating:** 6
**Confidence:** 3

**Summary:**

This paper proposes a neural architecture search (NAS) method for tabular data. The proposed method, termed ATLAS, leverages a training-free metric for multi-layer perceptron (MLP) to estimate the architecture performances at low cost. After the filtering phase using the training-free metric, it searches for a better architecture using the accurate training-based architecture evaluation. In the proposed two-phase search strategy, the switching timing is determined based on a given search budget to improve the anytime performance. The effectiveness of the proposed method is evaluated on the NAS problem of finding the best number of units in each layer of MLP.

**Strengths:**

- The NAS tabular data benchmark constructed in this work will be useful for the community. It would be great if the authors released the dataset publicly.
- The thoughtful experimental evaluation is conducted. The proposed training-free metric, ExpressFlow, empirically shows better correlations with the actual performance compared to several existing training-free metrics. Also, the anytime performance of the proposed method clearly outperforms the baselines.
- The paper is generally well-written and easy to follow.

**Weaknesses:**

- The performance evaluation is conducted using the limited search space that decides the number of hidden units in each layer of MLP. The effectiveness of the proposed method on other architecture search spaces, such as Transformer-based architectures and other components of MLP, is unclear.

**Questions:**

- Could you comment on the applicability and expected behavior of the proposed method on other kinds of architecture search spaces other than the search space used in this paper?
- Is it possible to apply the proposed method to a situation where the additional budget will be available after the algorithm starts?

----- After the rebuttal -----

Thank you for answering my question.

The authors' responses are convincing to me. I would be happy if the discussion regarding my questions were added to the revised paper.
I keep my score to the acceptance side.

---

> ### Author Response · Authors · 2023-11-22
>
> We thank you for your constructive comments and positive feedback. We would like to address your concerns below.
>
> > The NAS tabular data benchmark ... It would be great if the authors released the dataset publicly.
>
> **Response:** Thank you for appreciating the contribution of our NAS tabular data benchmark (NAS-Bench-Tabular). We will make it publicly available to facilitate further research and development in the area of NAS for tabular data.
>
> > The performance evaluation is conducted using the limited search space ... and other components of MLP, is unclear.
>
> **Response:** Existing studies show that deep neural networks (DNNs) can already achieve state-of-the-art performance on tabular data [1, 2], and the technical challenge is to configure DNNs with the right number of layers and hidden layer sizes for each layer [3]. Therefore, we design a DNN-based search space that comprises up to 160,000 candidate DNNs with an extensive collection of different layer configurations.
>
> As shown in the experiments and visualizations in Section 4.1.1 and Appendix C.3 of the original paper, this search space contains a diverse set of architectures with a wide range of parameter sizes, configurations, and performances, which captures the main properties of DNNs for benchmarking NAS approaches on tabular data.
>
> >Could you comment on the applicability ... search space used in this paper?
>
> **Response:** Our proposed method, ExpressFlow, is transferable and can be applied to other search spaces, such as NAS-BENCH-101 and NAS-BENCH-201 [4], which are designed for vision tasks.
>
> However, ExpressFlow is specifically tailored for tabular data, as it is based on neuron saliency that captures the complex and non-intuitive relationships among input features in this data type.
>
> Consequently, while demonstrating higher correlation values on tabular datasets (as shown in Table 1 of the original paper), ExpressFlow achieves only the second-highest average rank in terms of correlation value for image data, as illustrated in Table 15 of Appendix G.1.
>
> > Is it possible to apply the proposed method ... additional budget will be available after the algorithm starts?
>
> **Response:** Thank you for the insightful comments and suggestions. We agree that this is a very practical situation.
>
> Our NAS approach is anytime and thus can readily adapt to this situation by continuing to search for better-performing architectures with the current available time. Specifically, our approach includes two distinct phases, namely, the filtering phase and the refinement phase, along with a coordinator. If the additional budget is available during the filtering phase, the coordinator can dynamically allocate more time to both phases, which allows for exploring more architectures in the filtering phase and exploiting more architectures in the refinement phase. Conversely, if the additional budget is available during the refinement phase, all budgets are allocated to this phase, and thus each architecture will be trained over more epochs during the successive halving process.
>
> We hope our responses above have addressed your concerns and can improve your evaluation of our work.
>
> [1] Well-tuned simple nets excel on tabular datasets [Kadra et al., NeurIPS 2021].
>
> [2] TabNAS: Rejection Sampling for Neural Architecture Search on Tabular Datasets [Yang et al., NeurIPS 2022].
>
> [3] Revisiting Deep Learning Models for Tabular Data [Gorishniy et al., NeurIPS 2021].
>
> [4] Nas-bench-suite-zero: Accelerating research on zero cost proxies [Krishnakumar et al., NeurIPS 2022].

---

### Author Response · Authors · 2023-11-22

We thank the reviewers for your insightful and constructive comments on our paper. In this **global response**, we would like to address the common concerns.

> **Concern 1:** Evaluation of ATLAS on more Datasets.

**Response:** To validate the generalizability and effectiveness of our ATLAS, we have evaluated it on eight diverse datasets from the OpenML AutoML Benchmark. These datasets range from small (under 1,000 data samples) to large (10,000 data samples), with varying column numbers (5 to 2001). They encompass both binary and multi-class classification problems.

For each dataset, we have executed our ATLAS to identify a higher-performing architecture. Then, we have fully trained the selected architecture and evaluated its performance in terms of balanced accuracy.

To further illustrate the effectiveness of our ATLAS, we have also compared ATLAS against five baseline methods including unregularized MLP designed by [1] (MLP), XGBoost (XGB), CatBoost (CatB), Gradient-Boosted Decision Tree (GBDT) implemented by Auto-sklearn (ASK-G), and AutoGluon with stacking enabled (AutoGL.S).

The results are shown in the Table below. First, the architecture selected by our ATLAS consistently outperforms the MLP across all datasets. Second, ATLAS achieves comparable performance with XGBoost, CatBoost, Auto-sklearn, and AutoGluon.

The experimental results confirm the effectiveness of ATLAS across various tabular datasets with varying data samples, feature dimensions, complexity, and domain-specific challenges. Moreover, the experimental results align with conclusions drawn from [2]: specific tuning can make simple models like MLPs competitive. Notably, our ATLAS effectively determines the number of hidden neurons for each layer of an MLP, thereby making it competitive with other methods.

|        | **Dataset** | **#Samples/ #Columns/ #Class** | **Task Type** | **MLP** | **XGB** | **CatB** | **AutoGL.S** | **ASK-G** | **ATLAS** | **Searched Architecture** | **Total Cost (CPU Sec.)** |
| ------ | ----------- | ------------------------------ | ------------- | ------- | ------- | -------- | ------------ | --------- | --------- | ------------------------- | ------------------------- |
|        | dilbert     | 10000 / 2001 / 5               | Multiclass    | 96.930  | 99.106  | 99.259   | 98.704       | 98.758    | 99.041    | 512-512-512-256           | 136.7                     |
| Large  | christine   | 5418 / 1637 / 2                | Binclass      | 70.941  | 74.815  | 73.708   | 74.170       | 74.447    | 74.416    | 512-384-512-256           | 72.0                      |
|        | fabert      | 8237 / 801 / 7                 | Multiclass    | 63.707  | 70.098  | 71.708   | 68.142       | 70.120    | 68.651    | 256-384-512-512           | 24.0                      |
| Medium | jasmine     | 2984 / 145 / 2                 | Binclass      | 78.048  | 80.546  | 80.052   | 80.046       | 78.878    | 80.056    | 512-512-384-384           | 16.0                      |
|        | sylvine     | 5124 / 21 / 2                  | Binclass      | 93.070  | 95.509  | 95.119   | 93.753       | 95.119    | 95.046    | 512-256-512-256           | 14.0                      |
|        | car         | 1728 / 7 / 4                   | Multiclass    | 97.442  | 92.376  | 100.000  | 99.675       | 100.000   | 100.000   | 384-256-256-128           | 1.3                       |
| Small  | australian  | 690 / 15 / 2                   | Binclass      | 86.268  | 89.717  | 91.016   | 88.248       | 88.589    | 90.366    | 384-256-256-512           | 0.6                       |
|        | blood       | 748 / 5 / 2                    | Binclass      | 67.836  | 62.281  | 59.576   | 67.251       | 64.985    | 68.640    | 512-256-256-512           | 0.7                       |

>**Concern 2:** Reasons for Selecting the Three Datasets for NAS-Bench-Tabular

**Response:** We utilize those three datasets for two reasons: (1). They are well-known datasets covering various domains, including app recommendations, healthcare, and CTR prediction. (2) They represent diverse classification problems, containing between 101,766 and 45,840,617 data samples, and between 369 and 5,382 features, varying in terms of the number of numerical and categorical features.

The dataset statistics are summarized in the Table below.

| Dataset  | # Class | # Samples  | # Columns | # Features | Task                 |
| -------- | ------- | ---------- | --------- | ---------- | -------------------- |
| Frappe   | 2       | 288,609    | 10        | 5,382      | App Recommendation   |
| Diabetes | 2       | 101,766    | 43        | 369        | Healthcare Analytics |
| Criteo   | 2       | 45,840,617 | 39        | 2,086,936  | CTR Prediction       |


[1] Well-tuned simple nets excel on tabular datasets [Kadra et al., NeurIPS 2021].

[2] Revisiting Deep Learning Models for Tabular Data [Gorishniy et al., NeurIPS 2021].

---

### Meta-Review · Area_Chair_ckU8 · 2023-12-07

**Metareview:**

This paper studies training-free evaluations for NAS on tabular data, combined with standard successive halving on the highest-ranked architectures. The paper also introduces a novel NAS Benchmark on tabular data.
Strengths highlighted by the reviewers include the tailored NAS algorithm for tabular data, the new NAS benchmark, clarity of writing, and code availability.
Weaknesses highlighted by multiple reviewers include the limited search space (only parameterizing MLPs), narrow selection of tabular benchmarks, missing comparison to classic tabular data methods, and false claims of being the first NAS method with the anytime property.
(On top of what Reviewer uv1h wrote, the TPAMI 2021 paper "Auto-PyTorch Tabular: Multi-Fidelity MetaLearning for Efficient and Robust AutoDL" likewise uses successive halving, is efficient, and also introduces a benchmark for tabular data.).
In their rebuttal, the authors report results for some datasets of AutoML benchmark, but this evaluation does not state details on the experimental protocol used, and unfortunately, the devil is in the detail here: what versions of the baselines were used, with which hyperparameter search spaces, which hyperparameter optimization methods and which budget? There are far too many things to be done wrong here. E.g., I would expect AutoGluon 1.0 to be much better than the weak performance the authors reported for AutoGluon.
Overall, while the paper is very interesting, I believe it needs another iteration to properly situate it in the existing literature. I very much encourage the authors to continue this line of work.

**Justification For Why Not Higher Score:**

Limited search space (only parameterizing MLPs), narrow selection of tabular benchmarks; missing / limited comparison to classic tabular data methods.

**Justification For Why Not Lower Score:**

N/A

---

### Decision · Program_Chairs · 2024-01-16

Reject